# Satellite record reveals 1960s acceleration of Totten Ice Shelf in East Antarctica

Rongxing Li [1,2], Yuan Cheng [3] ✉, Tian Chang [1,2], David E. Gwyther [4], Martin Forbes[5], Lu An [1,2] ✉, Menglian Xia[1,2], Xiaohan Yuan[1,2], Gang Qiao[1,2], Xiaohua Tong[1,2] & Wenkai Ye[1,2]

Wilkes Land and Totten Glacier (TG) in East Antarctica (EA) have been losing ice mass significantly since 1989. There is a lack of knowledge of long-term mass balance in the region which hinders the estimation of its contribution to global sea level rise. Here we show that this acceleration trend in TG has occurred since the 1960s. We reconstruct ice flow velocity fields of 1963–1989 in TG from the first-generation satellite images of ARGON and Landsat-1&4, and build a five decade-long record of ice dynamics. We find a persistent long-term ice discharge rate of $68 \pm 1$ Gt/y and an acceleration of $0.17 \pm 0.02$ Gt/y$^2$ from 1963 to 2018, making TG the greatest contributor to global sea level rise in EA. We attribute the long-term acceleration near grounding line from 1963 to 2018 to basal melting likely induced by warm modified Circumpolar Deep Water. The speed up in shelf front during 1973–1989 was caused by a large calving front retreat. As the current trend continues, intensified monitoring in the TG region is recommended in the next decades.

The Antarctica Ice Sheet (AIS) has been losing ice mass and contributing to global sea level rise (SLR) at an accelerated pace over the last few decades[1]. This has been determined by estimates from altimetric, gravimetric, and optical and SAR imaging satellite observations, with the main mass loss from West Antarctica (WA) and the Antarctic Peninsula (AP)[1–3]. A reconciled solution shows a mass change rate with a large uncertainty ($5 \pm 46$ Gt/y) in East Antarctica (EA)[4]. However, a recent finding indicates that Wilkes Land sets itself apart from this overall trend in EA by showing a high mass loss rate of $51 \pm 13$ Gt/y (refs. [1], [5], [6]). Furthermore, studies[1,7] have noted that the Totten Glacier (TG), fed by the largest drainage basin in Wilkes Land[8], has undergone consistent mass loss since 1989, losing a total of ~175 Gt (or 0.5 mm SLR) at a rate of $7 \pm 2$ Gt/y from 1989 to 2015. Limited by the quality and availability of satellite data, there is generally a lack of mass balance estimates for the AIS and this region prior to 1989 (ref. [9]). Consequently, the lack of long-term mass balance knowledge hinders our ability to understand earlier glaciological responses in EA and the entire AIS to climate change.

Totten Glacier is marine based, mostly grounded below sea level and the bedrock slopes downwards away from the ocean[10]. Thus, the glacier is subject to marine ice sheet instability (MISI) and vulnerable to potential rapid ice sheet retreat due to the intrusion of the relatively warm modified Circumpolar Deep Water (mCDW) from the Southern Ocean[11–15]. With a catchment of ~570,000 km$^2$ (ref. [12]), the TG basin has the capacity to raise the global sea level by 3.9 m (ref. [16]) if completely melted. Its ice shelf has been experiencing increased basal melting at a rate from ~9.1 m/y (1992–2007) (ref. [13]) to ~18 m/y (2005–2011) (ref. [17]). The grounding line retreated up to ~3 km during 1996–2013 (ref. [16]). Between 1989 and 2015, the ice flow acceleration was found to be as large as 18% (refs. [7], [18]). However, a longer satellite observation record is needed to put the recent variability in a better context. Although recent satellite data of InSAR, altimetric and gravimetric sensors are not available before 1989, early optical images collected on

[1]Center for Spatial Information Science and Sustainable Development Applications, Tongji University, 1239 Siping Road, Shanghai, China. [2]College of Surveying and Geo-Informatics, Tongji University, 1239 Siping Road, Shanghai, China. [3]Institute for the Conservation of Cultural Heritage, School of Cultural Heritage and Information Management, Shanghai University, Shanghai, China. [4]School of Earth and Environmental Sciences, The University of Queensland, St Lucia, QLD 4072, Australia. [5]National School of Surveying, University of Otago, Dunedin, New Zealand. ✉e-mail: chengyuan@shu.edu.cn; anlu2021@tongji.edu.cn

films by the ARGON intelligence satellites during the 1960s (ref. [19]) and the Landsat satellites in the 1970s and 1980s can be used to recover historical glacier topography and ice flow velocity fields[9,20,21] with comparable uncertainties of earlier and recent velocity products[1,22,23]. Furthermore, these ice velocity data along with other supporting data will allow us to estimate mass balance during historical periods at both glacial and continental scales through the input–output (IO) method[1,5].

To quantify the early state of ice dynamics and mass balance in the TG region, we developed and applied an innovative method of hierarchical network densification to systematically map Antarctic ice velocity from historical optical satellite images. Here, we used images in the TG region acquired by the first-generation satellites of ARGON and early Landsat missions in order to reconstruct the ice velocity fields of multiple periods from 1963 to 1989. We found ice flow acceleration and high-level ice discharge during the period that are linked to a large calving front retreat during 1973–1985 and increased basal melting of the ice shelf. Finally, using our historical estimates alongside recent results, we built a nearly 60-year record and proved that there has been a persistent, long-term ice discharge increase driven by continued ice shelf basal melting over 6 decades in this marine-based region, making TG the glacier with the highest mass loss in EA. We suggest that the current acceleration and mass loss process in the TG region had already started by the 1960s.

## Results

### Reconstructed velocity maps

A historical velocity map of the TG region from 1963 to 1989 is generated (Fig. 1a), reconstructing the velocity field using 71,442 velocity vectors derived from ARGON and Landsat MSS and TM satellite images. This overall map is produced by weight-averaging three periodic velocity maps with shorter timespans (1963–1973, 1973–1989, and 1989) which are used to analyze ice flow dynamics and estimate the mass balance of the drainage basin over the 26-year period (Fig. S1). Displacements of the velocity vectors were computed by tracking ice flow features in image pairs using the innovative hierarchical matching and network densification method developed for glacier mapping using historical Antarctic images[21,23]. Unique processing techniques for overcoming difficulties inherent in the historical images were developed and applied, including film deformation, large format lens distortion, and georeferencing errors exceeding 10 km (refs. [21], [24–26]). For instance, we adopted a semi-automatic algorithm for recognition and measurement of damaged fiducial marks on ARGON films; exterior orientation parameters were initially estimated from the ephemeris data and ground features, and then refined through a bundle adjustment procedure. Consequently, uncertainties of the produced velocity maps range from 4 m to 79 m (Table S1), which were estimated from the orthorectification error of the images, feature identification and matching errors, and timespan according to the error propagation law described in the "Methods" section.

The long timespan velocity maps of 1963–1973 and 1973–1989 have the benefit of covering the grounded regions and most of the ice shelf at a low uncertainty of 4–17 m/y (Table S1), especially the low velocity areas (<10–80 m/y), such as the Law Dome (Figs. S1a, b and 1a). However, the fast-flow shelf front area (up to ~2300 m/y) cannot be mapped due to lost tracking features that were calved into the ocean over timespans of up to 16 years. This data gap was effectively filled by the 8-month velocity map of 1989 (Fig. S1c) at an uncertainty of 79 m/y. The overall velocity map of the TG from 1963 to 1989 with a grid spacing of 500 m (Fig. 1a) was generated by weight-averaging the velocity points from the three periodic maps within a natural neighbor[27,28]. The extended area (outside the gray dashed line in Fig. 1a) is covered by using a regional velocity map[29,30].

Overestimation (OE) of velocity due to ice flow acceleration was found in long timespan velocity maps (e.g., over 3 years)[9,31]. If not corrected, the implication is that the overestimated historical velocity, when combined with the recent unbiased velocity, may cause an underestimation of the long-term velocity change used to study climate impact on a decadal scale. We developed an innovative Lagrangian velocity-based method for OE correction without the use of field observations or additional image data[9]. The OEs in this region are mainly distributed along the main trunk, in the grounding zone, and at the shelf front (Fig. S2). The average of the OE corrections is 50 ± 39 m/y for the velocity maps of 1963–1973 and 1973–1989. The velocity map of 1989 did not need OE correction because of the short timespan of 8 months.

### Ice flow kinematics

The velocity of grounded ice in the slow-flowing area (<80 m/y, Fig. 1a) of the TG changed little, on average 6 ± 16 m/y (~0%), from 1963 to 1989 as indicated by the two velocity difference maps (Fig. 1b, c), indicating insignificant variation of discharge from the upper part of the grounded ice which would contribute to sea-level rise. However, acceleration was mainly found on the ice shelf, with an average velocity increase of 60 ± 11 m/y (~7%) from 1963–1973 to 1973–1989 for the floating ice. An increase of up to 135 ± 9 m/y (~10%) occurred near the shelf front during the same period. This shelf-wide acceleration was in the same time range as a large calving front retreat between 1973 and 1989 (Fig. 1a), with an area loss of ~645 km² (12%). We further narrowed the retreat period to 1973–1985 using images of the Advanced Very High-Resolution Radiometer (AVHRR), during which the retreat may have resulted from a major calving event or a series of smaller calving activities initiated by numerous rifts[32]. This retreat is found to be the largest for the Totten Ice Shelf (TIS) since the region was photographed by the earliest ARGON satellite in 1963. In particular, a large portion of the lost area crossed the "passive shelf-ice" (PSI) boundary[33] in the western shelf margin (Fig. 1a). Loss of this "active shelf-ice" area (104 km² or ~2%) caused velocity responses throughout the ice shelf[32,34]. Ice velocity on the ice shelf remained high in 1989, at least 4 years after the retreat, without significant changes from the 1973–1989 map (Fig. 1c). Furthermore, the acceleration from 1963–1973 to 1973–1989 is clear on the ice shelf and in a 20 km zone upstream and downstream from the grounding line along centerline AA′ (Fig. 1a, d). Similarly, velocity increased along centerline BB′ of the eastern tributary glacier (Fig. 1a) by up to 207 ± 55 m/y (~14%) near the shelf front (Fig. 1e). However, the velocity on grounded ice and near grounding line showed no significant changes, indicating no variations in discharge and direct contribution to sea-level rise from this part of the glacier. It is also suggested that the strong calving front retreat in the western margin may have not affected the ice dynamics in the tributary on the other side of the ice shelf.

To link the reconstructed ice dynamics during our study period with recent acceleration in the TG, we built a nearly 6-decade record of velocity in two locations, one near the shelf front and the other near grounding line (Boxes 1 and 2 in Fig. 1a), using our historical velocity maps and recent velocity products[35–37]. We calculate changes in percentage during our study period relative to the long-term average from 1963 to 2018. The velocity near the shelf front was initially high, at 1328 ± 9 m/y during 1963–1973 (Fig. 2a), which is 28 ± 19 m/y (~2%) above the long-term average velocity. Subsequent acceleration due to the large calving front retreat caused an increase of 113 ± 9 m/y (~11%), making the velocity of 1973–1989 the highest over the 55 years. After 1989, the velocity fluctuated mostly within a range of ~50 m/y. Note that the acceleration was, therefore, not constant near the shelf front throughout the time period. The long-term ice velocity trend (Fig. 2a) appears to be consistent with that of the area loss and gain caused by calving front retreat and advance (Fig. 2b), with $R^2 = 0.8$ estimated between the two time series using timespan as weights (Table S3). Therefore, we suggest that the long-term ice velocity trend near the shelf front of TIS has mainly been controlled by calving activities.

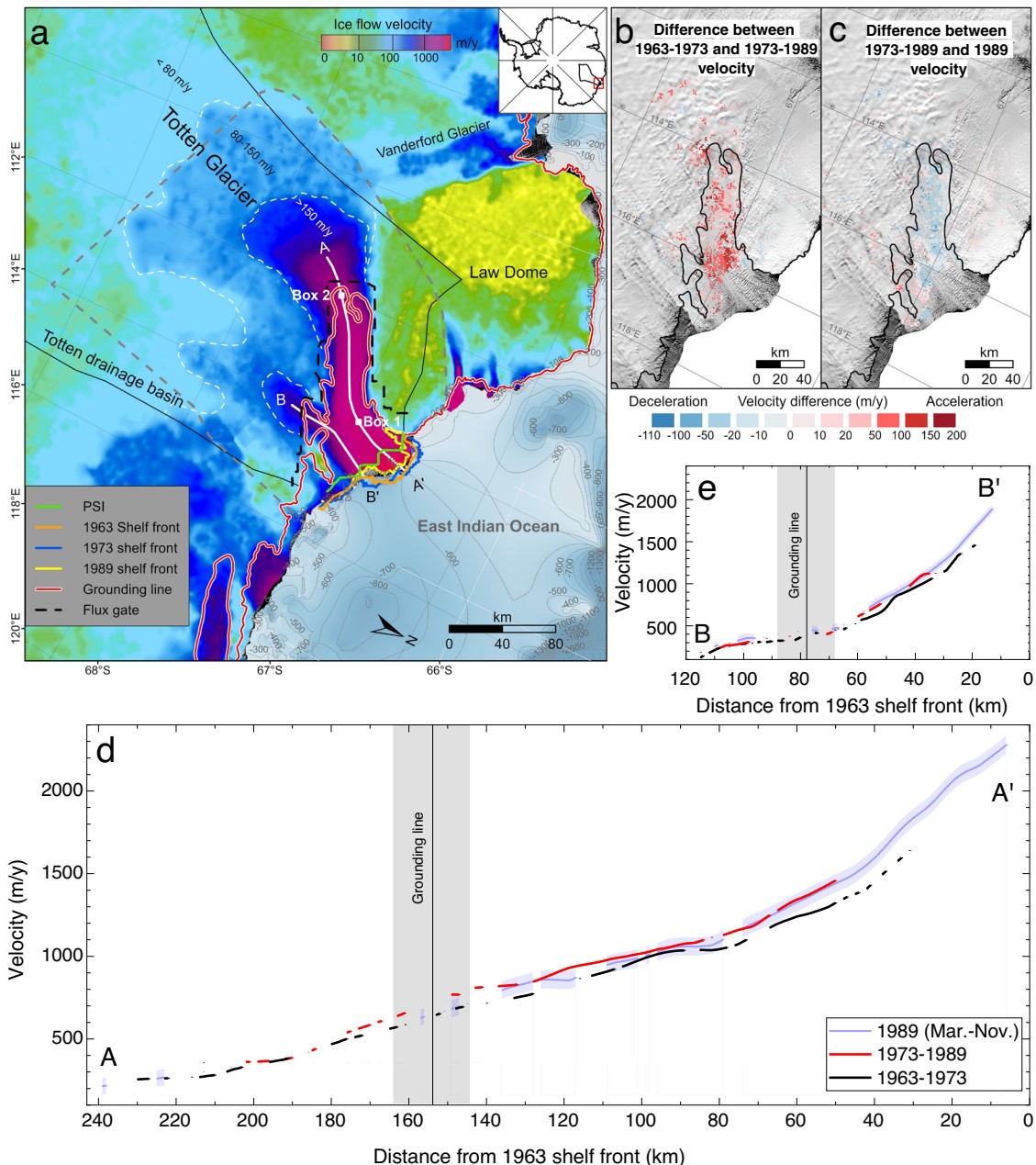

**Fig. 1 | Reconstructed historical ice velocity record revealed an active ice dynamic state in the Totten Glacier (TG) region. a** Ice flow velocity field from 1963 to 1989 with glacier and ice shelf centerlines marked AA' and BB', respectively, Box 1 in shelf front (-50 km from shelf front), Box 2 near grounding line (-4 km from grounding line) and shelf fronts showing the largest retreat during 1973–1989. The gray dashed line is the dividing line between the velocity maps generated in this study and the regional velocity map[29, 30]. **b** Velocity difference map (1973–1989 minus 1963–1973), illustrating an ice shelf-wide increase of $60 \pm 11$ m/y, due to the shelf front retreat. Red denotes acceleration and blue denotes deceleration. **c** Velocity difference map (1989 minus 1973–1989), showing that the ice velocity in 1989 remained high, with a difference of only $-15 \pm 55$ m/y. **d** Ice velocity changes from 1963 to 1989 along centerline AA' of the main trunk of TG. **e** Ice velocity changes from 1963 to 1989 along centerline BB' of the eastern tributary glacier. The backgrounds of **a**–**c** are from the Landsat image mosaic of Antarctica (LIMA mosaic)[69]. The ice shelf fronts in **a** are digitized from the ARGON and Landsat orthoimages. The grounding line is from ref. 70. The Passive Shelf Ice (PSI) boundary (green line) in **a** is from ref. 33. Bathymetry data (100 m isobath) on the continental shelf in **a** are from ref. 59. Source data are provided in the Source Data file.

The ice velocity during our study period was historically high due to the largest calving front retreat between 1973 and 1985.

Conversely, the velocity near grounding line (Box 2 in Fig. 1a) was initially low during 1963–1973, $123 \pm 21$ m/y (-15%) below the long-term average velocity (Fig. 2c). There is a long-term increasing trend of velocity over the period of 1963–2018 which is consistent with that of the modeled melt rate from 1960 to 2007 (ref. 38), simulated with COREv2 forcing[39] (Fig. 2d). The simulated melt rate shows lower basal

melting of -5 to 6 m/y. Thereafter melt rate estimated from multi-mission altimetry observations increased to $11.5 \pm 2$ m/y from 1994 to 2018 (ref. 40) and $17.9 \pm 1.2$ m/y from 2005 to 2011 (ref. 17). We reconstructed a velocity time series from that in Fig. 2c by subtracting its long-term linear trend (black dashed line) estimated by a regression, which then shows a timespan weighted correlation of $R^2 = 0.6$ with the shelf front area change in Fig. 2b. Thus, the acceleration during 1973–1989 in Fig. 2c, with a velocity increase of $124 \pm 8$ m/y (-0%),

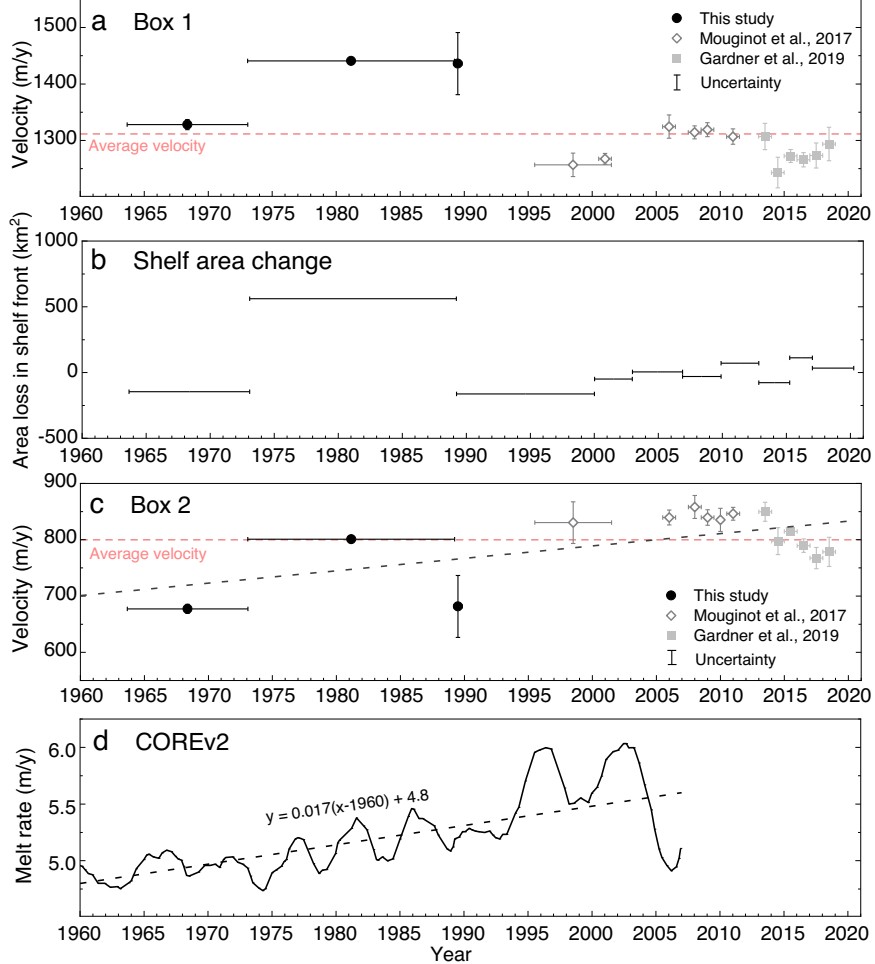

**Fig. 2 | Long-term ice velocity trend in Totten Ice Shelf (TIS) from 1963 to 2018.**
**a** Ice velocity near the shelf front (Box 1 in Fig. 1a) during the study period of 1963–1989 shows acceleration caused by the largest calving front retreat and the highest velocity in nearly 6 decades; **b** shelf area change (loss or gain) due to calving front retreat or advance indicates correlation with the ice velocity trend near the shelf front in **a** ($R^2 = 0.8$); **c** ice velocity near grounding line (Box 2 in Fig. 1a) from 1963 to 2018 shows a long-term increasing trend (black dashed line) consistent with basal melt rate in **d**, in addition to the acceleration induced by the calving front retreat during 1973–1989; **d** simulated area-averaged basal melt rate of TIS from 1960 to 2007 (ref. [38]) with an increasing trend (black dashed line). Source data are provided in the Source Data file.

appears to be caused by the largest calving front retreat during the same period, whose area loss intruded into the "active" shelf ice region inside the PSI boundary and influenced ice kinematics in the grounding zone. The ice velocity declines as elsewhere on the ice shelf in 1989, but then picks back up again. This recent acceleration after 1989, also reported in refs. [1], [5], [6], [16], was induced by ice shelf basal melting caused by intrusion of mCDW from the continental shelf into the ice shelf cavity through the trough on the seafloor[7,8,41]. The combined long-term melt rates from 1960 to 2018 based on modeling and satellite observations suggest that ice shelf basal melting may have existed in TIS as early as 1960. Here we demonstrate that the long-term acceleration trend in the grounding region induced mainly by ice shelf basal melting (Fig. 2c) started in 1963, 26 years earlier than reported.

**Mass balance**

Due to the ice flow acceleration in the grounding zone from 1963 to 1989 caused by both calving front retreat and basal melting, ice discharge in the TG was at a high level, close to the reference surface mass balance (SMB) that is the long-term average SMB from 1979 to 2016 (Fig. 3a). Consequently, a total of 1774 ± 16 Gt (66 ± 1 Gt/y) ice mass was discharged across the grounding line during the period (Fig. 3b). However, SMB in this period was higher, on average ~3 Gt/y above the reference SMB (Fig. 3a). The excessive cumulative SMB of 1904 ± 22 Gt

(71 ± 1 Gt/y) in the TG basin overrode the discharge and resulted in a total net mass gain of 130 ± 27 Gt (5 ± 1 Gt/y) (Fig. 3b). However, as the SMB decreased and discharge increased in the TG basin, the net mass gain reduced to about zero at the end of the period (Table 1). Therefore, during 1963–1989, the TG basin experienced a high level of ice discharge, transiting from positive to negative mass balance.

## Discussion

Bindoff et al.[42] and Williams et al.[43] observed mCDW on the continental shelf adjacent to TIS in the austral summer of 1995–1996 and austral winter of 2007, respectively. Intrusion of the on-shelf warm mCDW water towards the TIS resulted in circulation of the cyclonic gyre and westward Antarctic Slope Current[42,44], modulated by polynya activities near the calving front[45]. A major entrance, a ~5 km wide trough, which allows the mCDW water access to the ice shelf cavity, was found in the western part of the TIS calving front using inverted bathymetry from airborne gravity and magnetic data collected between 2008 and 2012 (ref. [8]). The trough entrance was more precisely described, by shipboard bathymetry data acquired in January 2015, as being ~10 km wide, ~600 m deep, and at a depth of ~1097 m (maximum) from the sea surface[41]. Li et al.[7] suggested that ice flow acceleration in the TG between 1989 and 2015 was linked to ocean temperature changes. Furthermore, Greene et al.[14] and Gwyther et al.[38] indicated that the

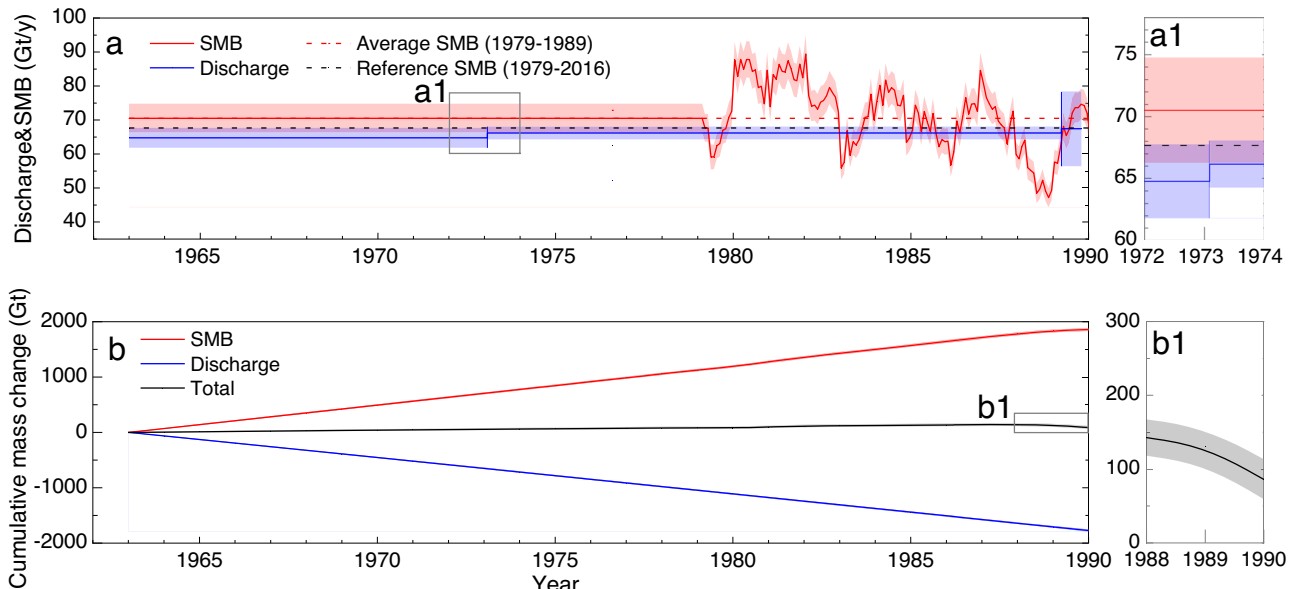

**Fig. 3 | Surface mass balance (SMB), ice discharge, and mass balance of Totten Glacier (TG) during 1963–1989. a** Ice discharge with uncertainty (blue line and shaded margin) and SMB from RACMO2.3 p2 (red line and shaded margin) in the TG basin during 1963–1989. Reference SMB (black dashed line) is averaged during 1979–2016. Average SMB (red dashed line) is calculated over the period of 1979–1989. Zoom inset "a1" shows details for 1972–1974. **b** Cumulative discharge (blue line), cumulative SMB (red line) and total mass change (net mass gain, black line) for the same period. Zoom inset "b1" shows net mass gain with uncertainty (shaded margin) of the last two years. Source data are provided in the Source Data file.

troughs underneath the ice shelf provided effective pathways for the mCDW water to intrude into the cavity and grounding zone, causing increased ice shelf basal melting and thinning at a rate from ~9 m/y for 1992–2008 (refs. 13, 46) to ~18 m/y for 2005–2011 (ref. 17).

Despite the spatial and temporal sparsity of oceanographic observations in our study area, we are able to use six conductivity, temperature, and depth (CTD) profiles (IDs 1–6) in an extended region (Figure S3), mostly collected during austral summer of 1995–1996 (ref. 42), from the World Ocean Database (WOD) supported by the NOAA Climate and Global Change Program (https://www.ncei.noaa.gov/). Limited by fast ice and grounded icebergs, the CTD profiles were not deployed close to the shelf front. They reveal the presence of mCDW of relatively high temperature (>~0 °C) and salinity (>~34.5 PSU) at ~300–600 m depth, near the continental shelf break and in front of TIS (Fig. S3b, c). We further performed a regional COREv2 simulation from 1960 to 2007 (Fig. S3d) and show that the shelf-wide pattern (Fig. 2d) is primarily attributed to the high melt signals in the grounding line region (~11 m/y) and margin regions (~5 m/y; Fig. S3e) where ice velocity and ice-flux can be altered. The melt rate in the shelf front region is low at ~2 m/y. Given the same bathymetry of the continental shelf, and the entrances to and troughs underneath the ice shelf cavity, we suggest that during our study period warm mCDW water may have crossed the continental shelf and intruded into the cavity, resulting in increased basal melting and changing in driving stress at the grounding line or resistive stresses along shear margins[47–51]. Therefore, the mCDW-induced basal melting process is likely responsible for the acceleration during our study period from 1963 to 1989 and further for the long-term acceleration trend of nearly 6 decades in the grounding line region of the TG (Fig. 2c).

Our ice shelf modeling results ("Method" section and Fig. S6) reveal that the shelf front retreat during 1973–1985, which caused loss of ice shelf contact with a long near-front section of western margin, induced significant speed up (>300 m/y) of shelf flow in proximity to the western margin (Fig. S6F). This instantaneous response explains the observed acceleration detected in Box 1 using the velocity maps

(Fig. 2a). After that, margin contact is regained; ice speed declines and then remains around the long-term average. Furthermore, the modified boundary conditions due to the shelf front retreat impacted the entire ice shelf and immediate grounded areas of grounding line with an average velocity increase of ~84 m/y (Fig. S6F). This explains the observed shelf-wide acceleration from 1963–1973 to 1973–1989 (Fig. 1b). The modeled velocity response in the grounding line region to the calving front retreat is <10m/y, lower than the observed increase in Box 2 (Fig. 2c).

On the decadal scale, the mCDW-induced basal melting indirectly drove the persistent ice discharge increase from 1963 to 2018 (Fig. S4a). The TG basin discharged ice mass under the long-term average SMB (reference SMB) until reaching an equilibrium point in 1989. After this transition, the glacier flowed at speeds above that required to maintain a state of mass balance. TG discharged a total of 3830 ± 31 Gt over the entire period from 1963 to 2018. Using the transition point of 1989 as the starting time and reference SMB (67.7 ± 4.1 Gt/y) as the basis for forward and backward cumulative mass change computation, we demonstrate that TG discharged 66 ± 1 Gt/y on average during the first 27 years, compensating for the high SMB by an additional amount of 54 ± 12 Gt (2 ± 1 Gt/y) and reaching equilibrium in 1989 (Fig. S4b). Although we cannot show short timespan (e.g., annual) variations, the overall discharge increase trend during the period is clearly demonstrated. Another 92 ± 28 Gt was discharged excessively in the last 29 years (3 ± 1 Gt/y) to accelerate mass loss. The overall ice discharge acceleration during the 56 years is 0.17 ± 0.02 Gt/y². The cumulative mass balance of the TG basin was dominated by ice discharge, but it was modulated by SMB ranging from -26 Gt to 27 Gt and two significant decreases during 1980–1989 and 2005–2018. Over the entire period of more than 5 decades, the cumulative mass balance in the TG basin changed from a net mass gain of 130 ± 27 Gt during 1963–1989 to a net mass loss of 136 ± 36 Gt during 1990–2018.

TG is a marine-based glacier[16] and has the eastern lobe of its grounding line along the main trunk, sitting on a retrograded bed, away from the ocean (Fig. S3d). Thus, it is subject to marine ice sheet

**Table 1 | Mass balance (MB) estimation from three velocity maps**

| Ice velocity map | SMB (Gt/y) | Ice Flux (Gt/y) | | | MB (Gt/y) |
|---|---|---|---|---|---|
| | | F | dM$_{FG}$ | D | |
| 1963–1973 | 67.7 ± 4.1 | 62.5 ± 3.0 | 2.2 ± 0.1 | 64.7 ± 3.0 | 3.0 ± 5.0 |
| 1973–1989 | 69.5 ± 4.2 | 63.9 ± 1.9 | 2.3 ± 0.1 | 66.1 ± 1.9 | 3.4 ± 4.6 |
| 1989 | 67.1 ± 4.0 | 65.0 ± 10.9 | 2.5 ± 0.1 | 67.4 ± 10.9 | -0.4 ± 11.6 |

*F is ice flux across the flux gate (Fig. 1a). dM$_{FG}$ is the correction applied to F. D is ice discharge across the grounding line (D = F + dM$_{FG}$). MB is mass change rate (MB = SMB−D).*

instability (MISI) and has been retreating faster than the western lobe, located on a forward slope. Given the current grounding line position, the retreat rate of 0.15 km/y (refs. 16, 52) and the bed topography, we suggest that the eastern lobe was at the beginning of the current retroslope during our study period (1963–1989). The mCDW-induced grounding line retreat[16] may have begun as early as the 1960s. We further suggest that the TG is currently experiencing a process that occurred earlier at Pine Island Glacier, another marine-based and rapidly changing glacier on the AIS[53–55] where basal melting induced grounding line retreat and speedup occurred for several decades prior to sustained large scale shelf front retreat was observed since 2017[56]. Analogously, since 1990, the observed TIS calving front positions have been changing modestly, mostly between the shelf fronts of 1973 and 1989 (Fig. S5), and likely with influence from sea ice dynamics[14]. Meanwhile the mCDW-induced basal melting caused grounding line retreat and acceleration in the TIS for over three decades. Therefore, more rapid calving activities may be expected as basal melting persistently weakens the stability of the ice shelf, resulting in an imbalanced shelf front retreat.

Using historical images of the first-generation satellites of ARGON and Landsat-1 and -4, we reconstructed velocity fields in the TG region, EA from 1963 to 1989. The developed velocity mapping techniques effectively handled the lower quality satellite images and corrected the velocity overestimates caused by the combined effects of ice flow acceleration and long timespan between image pairs, which would have otherwise introduced underestimated long-term velocity changes and mass balance. We found an ice shelf-wide velocity increase trend during 1963–1989. The acceleration in ice shelf front for 1973–1989 is attributed to the large calving front retreat, while that in the grounding line region is mainly caused by the mCDW-induced basal melting, indirectly leading to an increase in ice discharge. The reconciled satellite observations and modeling results reveal that the continued basal melting in TIS drove the long-term acceleration near grounding line and associated ice discharge over the period from 1963 to 2018. With the results of the accelerated ice discharge and decreased SMB, the mass balance state of the TG changed from a mass gain of 5 ± 1 Gt/y during 1963–1989 to a mass loss of −5 ± 2 Gt/y during 1989–2018. We suggest that recently reported ice flow acceleration and the mass loss trend in the TG basin and the Wilkes Land sector of East Antarctica since the 1980s may have started in the 1960s. As this trend continues, intensified monitoring of ice-air-water interactions in the TIS region is recommended in the next decades.

## Methods

**Velocity map uncertainty.** We estimate velocity map uncertainty $\sigma_{vel}$ from the orthorectification error $\sigma_{ortho}$, feature identification error $\sigma_{ident}$, matching error $\sigma_{match}$, and timespan $\Delta t$ : $\sigma_{vel} = \frac{1}{\Delta t}\sqrt{\sigma_{ortho}^2 + \sigma_{ident}^2 + \sigma_{match}^2}$. $\sigma_{ortho}$ is calculated using the displacements of check points between the known positions and those measured from the ARGON and Landsat orthoimages (Table S1). $\sigma_{ident}$ is the error for identifying a feature on the reference (first) image and is set as 0.5 pixels. $\sigma_{match}$ is also set as 0.5 pixels. We used 2 pixels for

ARGON $\sigma_{ident}$ and 1 pixel for MSS $\sigma_{match}$ in the 1963–1973 maps because of their lower image quality. The computed uncertainties of the three velocity maps are listed in Table S1. Details of the uncertainty estimation method are presented in ref. 23.

**Mass balance.** Since RACMO2.3 p2 data are available after 1979, we used the average SMB of 1979–2016 (67.7 ± 4.1 Gt/y) as the long-term reference SMB. We used the reference SMB to fill the data gap of 1963–1978 (Fig. 3a). SMB uncertainties are estimated according to the elevation SMB bias scheme[7,57,58]. To compute the ice flux, we used the ice thickness dataset BedMachine Antarctica[59]. The flux gate (thick black dashed line in Fig. 1a) is set based on FG2 in ref. 5, but improved by moving toward locations with ice thickness errors close to 30 m[59]. The average ice thickness uncertainty in this region is reduced from 61 m along FG2 to 32 m along the improved flux gate. Furthermore, an adjustment computed as the SMB between the flux gate and grounding line using RACMO2.3 p2 is applied to obtain the flux (discharge) across the grounding line. The MB of the TG basin is finally estimated for each velocity map period by subtracting the discharge from the SMB (Table 1)[5,23]. The accumulative discharge, SMB, and MB (Fig. 3b) are calculated, with regards to the reference SMB, as an integration of each variable from 1963 to 1989.

**Basal melting 1960–2007.** The TG is simulated with the Regional Ocean Modelling System[60] framework, which has modifications to include ice-ocean interaction (following refs. 61, 62) using the three-equation parameterization[63]. The model domain was from 104.5°E–130°E with 1/15° resolution and 60°S–68°S with 1/30° resolution. Model bathymetry is based on RTopo[64] but with the addition of the Totten ice shelf cavity including the western trough cavity entrance[8]. Lateral boundary forcing and wind forcing are from the COREv2 1949–2007 reanalysis dataset[39], while surface heat and salt fluxes are composed of a combination of SSM/I sea ice formation observations[65] with COREv2 evaporation minus precipitation fields. The combination allows for long-term estimates of polynya activity and sea ice formation. Further information is given in ref. 38.

**Calving modeling.** We used Ice Sheet System Model (ISSM)[66], a finite element model, to simulate the velocity response of the Totten glacier and ice shelf to calving front retreat. This model is created using velocity and geometry prior to 1973 (Fig. S6). The model construction procedure includes inverting for an ice rigidity parameter, B, within the shelf and a friction coefficient, C, for grounded ice through a minimization of the difference between observed ice flow velocity of 1963–1973 and modeled velocity (Fig. S6B, C). The fidelity of the inferred basal friction structure, common to both domains, is not of interest so long as the ice flow is well described at larger (10s of ice-thicknesses). Consequently, the friction coefficient is obtained during different minimization iterations, without the use of a regularization term. Regularization is included and calibrated with an L-curve analysis[67,68], for the inversion of ice rigidity within the shelf where fine scale variations may impact the analysis outcomes (Fig. S6A). Finally, the model was used to simulate the velocity of 1985 with the 1985 shelf front (Fig. S6E) to test the model response to the calving front retreat during 1973–1985.

## Data availability

All of the data used in this study are available online: ARGON, Landsat and LIMA images from the United States Geological Survey (USGS) (https://earthexplorer.usgs.gov/); RACMO 2.3 p2 from Institute for Marine and Atmospheric Research, Utrecht University (https://www.projects.science.uu.nl/iceclimate/publications/data/2018/); MEaSUREs BedMachine Antarctica, Version 2 from NASA National Snow and Ice Data Center (NSIDC) (https://nsidc.org/data/NSIDC-0756/versions/2). Velocity maps generated in this study have been deposited in the Dryad database (https://datadryad.org/stash/share/qJNkqLJ5acd1pKeka5u6jFV5cxg5g4XL6OW8GfREN6E). Source data

are provided in the Source Data file of this paper. Source data are provided with this paper.

## Code availability
Code for velocity overestimation calculation is available from corresponding author upon request.

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

## Acknowledgements

We appreciate constructive comments and suggestions from referees and editors. This research has been supported by the National Key Research and Development Program of China (No. 2017YFA0603100, R.L., G.Q., and X.T.; 2021YFB3900105, L.A. and G.Q.), the National Science Foundation of China (41730102, R.L.), and the Fundamental Research Funds for the Central Universities (R.L., X.T., L.A., and G.Q.).

## Author contributions

R.L. designed and lead research; Y.C., T.C., D.E.G., M.F., L.A., M.X., Y.H., and W.Y. performed research; R.L., Y.C., D.E.G., M.F., L.A., G.Q., and X.T. analyzed data; and R.L. wrote the paper.

## Competing interests

The authors declare no competing interests.
