## [Peer Review File · Nature Communications]

Satellite record reveals 1960s acceleration of Totten Ice Shelf, East AntarcticaREVIEWER COMMENTS

Reviewer #1 (Remarks to the Author):

Using satellite data, the authors estimated the ice velocity of Totten Glacier from the 1960s. The authors present estimated historical glaciological changes and past ocean observations. This kind of work investigating the history of glaciological and oceanographic changes is valuable. I recommend that this manuscript for published in Nature Communications after revision. However, I have a few major comments about their oceanographic analysis. As I am an oceanographer, I only comment on the ocean part of this work.

Major comment

Line 232: Authors state that mCDW was observed in front of the TIS. I downloaded the data from WOD but I could not find the same observations described in Fig S3. I found similar data in 110W (not 110E). I suspect that authors seem to flip the sign of longitude and they may be plotting the observations from other parts of the Southern Ocean. If my guess is correct, I recommend authors remove Fig. S3 and revise ocean related part of their text. Based on my understanding, I do not think no oceanographic observations exist in front of the TIS before the 1980s.

Minor comment

Line 228: Grounded icebergs?

Line 224-237: If these are the correct datasets, it would be great if you could list the names of oceanographic campaigns and how they obtained these measurements. Are these ship-based CTD casts?

Line 287: As mCDW-induced basal melting ...: See my comment above. Authors may need to remove all ocean-related discussions if observations are located in the wrong places.

Reviewer #2 (Remarks to the Author):

Review of Li et al., Nature Communications

Satellite record reveals 1960s initiation of acceleration in Totten Glacier, Antarctica

General Comments

Li et al. utilise historic Landsat and ARGON optical imagery to track ice flow with a previously published hierarchical matching and network densification method. A novel contribution to the field is made as ice velocity of Totten Glacier, East Antarctica, is calculated between 1963 and 1989 for the first time, thereby significantly extending the time series of velocity observations available here.

Velocity measurements reveal an acceleration between 1963 and 1989 on Totten Ice Shelf both near the grounding line and near the ice front, but little change is observed on the grounded portion of Totten Glacier. In the context of the complete time series (extending to 2018), velocity peaked in 1989 near the ice front, but accelerated consistently near the grounding line, so long-term acceleration is only clear immediately seaward of the grounding line, and not at the shelf front. The authors attribute the long-term acceleration to the presence of warm Circumpolar Deep Water, and the associated shelf area loss and basal melt rate.

The manuscript would be improved through consideration of the following points:

1. The methodology is sound, although further transparency in error reporting is recommended in some instances (see specific comments on L60, 79 & 89) given unavoidable issues in the quality of historical imagery which gives rise to significant errors, particularly in the case of the ice velocity in 1989.
2. More spatial nuance is required in the interpretation of ice dynamics in the extended velocity time series. Interpretation should clearly distinguish the spatial variation in long-term acceleration trends across Totten Glacier and its ice shelf i.e. long-term acceleration occurs near the grounding line, but not near the shelf front. The lack of initial or long-term acceleration on the grounded ice should also be emphasised, given that discharge of grounded ice directly impacts Antarctica's contribution to sea-level rise, unlike the varying discharge of the floating portion of Totten Ice Shelf.
3. More clarity on the cause(s) of acceleration is required in the discussion. Good evidence, both in situ and modelled data, is provided to describe the behaviour of both ice shelf area and basal melt rates, but discussion of these explanations requires spatial nuance and further statistical confirmation (see specific comments on L174 & 183). Does shelf area have a statistical relationship with velocity observations near the grounding line (Box 2), as well as near the shelf front (Box 1)? Does the basal melt rate have a statistical relationship with velocity observations in both locations?

Reproducibility of the study is good due to the extensive material included in the Supplements. Reference to the literature is detailed and comprehensive throughout, which contextualises the findings nicely.

Specific Comments

L1: Results show velocity acceleration dating back to 1963, but do not prove acceleration began at this time. Acceleration prior to 1963 is feasible. Discussion focussed on acceleration observed on Totten Ice Shelf, rather than the grounded glacier. Suggest reword title to: 'Satellite record reveals 1960s acceleration of Totten Ice Shelf, East Antarctica'

L19-20: Reword for clarity. '... hinders the mass change forecasting and estimations of its contribution to global sea-level rise.'

L21: Is initiation of acceleration shown? Acceleration prior to 1963 isn't disproven in the study.

L28: Acceleration attributed shelf area loss and basal melt rates in text. Both should be mentioned explicitly here.

L34: The inclusion of references would be beneficial here.

L37: The uncertainty associated with this figure should be emphasised here.

L40: Expand on the meaning of 'important' in this sentence. Reference(s) should also be included.

L45: How will understanding glaciological responses in the 1960s help forecast to future contributions to sea-level rise, when observations from the 1990s are already available?

L47: Is the runaway effect associated with MISI already in effect here given the long-term rate of acceleration is not constant according to Figure 2?

L48: '... intrusion of the relatively warm mCDW...'

L51: Repetition from L40?

L55-58: Sentence needs rewording for clarity. Why exactly are velocity observations prior to 1989 necessary? How does this relate to the behaviour at Pine Island and Thwaites?

L58: Reword to 'recent satellite data', rather than newer.

L60: Do the earlier optical films yield velocity data with comparable errors to modern techniques?

L62: Sentence does not clearly relate to the rest of the paragraph.

L79: The creation of the overall velocity map (1963-1989) by averaging three periodic velocity maps should be stated explicitly here (in addition to in L96).

L79: Brief mention of error calculations is required in this paragraph, in addition to the information included in the Materials and Methods section and Table S1.

Figure 1A: White text is difficult to read, and grounding line colour is difficult to distinguish from the colour depicting fast-flowing velocity.

L86: A brief description of these processing techniques is required here.

L89: What is the error associated with these velocity fields?

L91: How are these accuracy values calculated?

L98: It would be useful to mark this extended area in Figure 1.

L101: Should read: 'If not corrected, the implication is that the overestimated historical velocity...?'

L126: The observation that velocity over grounded ice changed very little should be emphasised given that dynamic variation on grounded ice contributes directly to sea-level rise, unlike floating ice.

L127: Figure 1B & D shows acceleration on the fast-flowing grounded glacier ice was also minimal.

Figure 1B&C: Larger font required for title. Change titles to 'Difference between 1963-1973 and 1973-1989 velocity' and 'Difference between 1973-1989 and 1989 velocity'. For clarity, note in legend that red denotes acceleration, while blue denotes deceleration.

L136: Include PSI boundary on Figure 1A.

L138: Note that the loss of buttressing potential is unlikely to be the case because the velocity of the grounded ice was minimally affected.

L140: Error too large to discuss this result with confidence.

Figure 1E: Text in legend is partially cropped. How has the width of the grounding zone been determined?

L142: Figure 1D should be referred to in the text prior to Figure 1E.

L144: Emphasis should be placed on this result given that changes to grounded ice directly contribute to sea level rise, and not changes in floating ice.

Figure 2B: Change title to 'Shelf area change'

L165: Note that the acceleration is, therefore, not constant near the shelf front throughout the time period.

L168: How are calving activities related to the presence of CDW, if at all?

L174: Replace 'retread' with 'retreat'?

L174: Can the R2 between velocity increase and calving front retreat be calculated here?

L183: Can the R2 between basal melt rates and velocity observations be calculated?

Line 188: A discharge increase is not certain due to overlapping error margins (Figure 3A) so this cannot be stated with certainty.

L190: The fact that the surface mass balance is above the reference surface mass balance is not clear from Figure 3A. A zoom inset would perhaps be a useful addition.

Figure 3A: Include full y axis label

Figure 3B: The positive gradient of the total net mass (black line) is not clear. Again, a zoom inset or similar would perhaps be a useful addition here.

Table 1/Figure 3: Figure captions need to be more informative throughout, stating what each of the coloured lines represent, for example, and not include any interpretation of results.

How do Table 1 and Figure 3B align? The straight lines in Figure 3B are not representative of the varying SMB/Discharge rates displayed in Table 1.

Line 234: Is there any evidence that the CDW reaches the grounding zone?

L236: Are you able to separate the roles of basal melt vs. front retreat? Do they have different influences at different locations on Totten Glacier and its ice shelf?

L259: Is there evidence or a reference for grounding line retreat here?

L260: Expand upon what is occurring at Pine Island Glacier, and how the evidence at Totten Glacier aligns.

Discussion on the role of calving front retreat also required given its influence on velocities near the ice front, demonstrated in Figure 2B.

L282: Spatial nuance required here. Long-term acceleration only observed in Box 2 (near grounding line), not Box 1 (near shelf front).

Reviewer #3 (Remarks to the Author):

Key Results

The manuscript reports on new insights developed using velocity from historical image materials that extend the observational record of Totten Glacier flow much earlier than previously available.

On my view, the data support the following claims

- 1) There is an overall trend toward negative net mass balance that began earlier than previously recognised.
- 2) A shelf-wide speed up was coincident with calving that would have reduced drag provided by the left margin (Figure 1, Figure 2, Figure S5).
- 3) While the shelf front speed reduced toward the long-term average after contact was regained with the left front margin (Figure 2, Figure S5), the grounding zone appears to have experienced a more persistent speed up despite that, requiring an additional explanation.

This leads to new insight. Conclusions in the literature (shown in Figure 2), that velocity has a multi-year cycle of speed up and slow down associated with intrinsic variability, are limited by the short time span they cover. The new data presented here shows us that the higher frequency variability is superimposed on other time scales of variation, some of which appears to be a direct result of calving and some of which requires further explanation.

The recent velocity data (Figure 2) exhibits variability with time scales similar to those in basal melting predicted by an ocean cavity circulation model, making ocean heat in the ice shelf cavity a likely driver for change that is unlikely to be explained by changes near the shelf front.

The Totten Glacier basin mass balance calculations produce another meaningful result, implying a longer term trend toward negative mass balance than previously concluded. Speculation about what may come next is overstated relative to the data presented here.

Validity

The paper presents glacier kinematics associated circumstances. Contrary to several statements in the text, the “state of ice dynamics of the TG region” has not been established. Possible causes for the observed change have not been evaluated quantitatively (that is, using models or mechanical analysis of the observations). The circumstantial case (involving the new observations and published simulations of ice shelf cavity circulation and basal melting) is not made as completely as it could be due to missing glaciological reasoning.

Significance

The new observations presented here are meaningful because they put recent change into a more climatologically relevant context, and because they offer more evidence with which to evaluate possible causes for the observed change. The longer time series challenges some recent focus on intrinsic variability only, showing that when the longer view is available, that variability may in fact be superimposed on a trend.

Data and methodology

The methods applied here are established and reviewed in the literature.

The “study period” is the time interval of the new data (1963 to 1989) but other ranges are used in some figures. This makes sense for Figures 1 and 2. But why does Figure S4 end in 2015 when the data in Figure 2 extend to 2018?

Figure 4S is a bit misleading (unintentionally so, I think). As plotted, the early part of the record looks like a steady decline toward negative MB but the early data are average representations of intervals that must surely have experienced variability of the sort shown in the more recent part of the record.

Analytical approach

Claims regarding causation are not as well supported as they should be (or could be). In the Conclusion, “We found an ice velocity increase trend during 1963–1989, which is attributed to the combined effect of the mCDW-induced basal melting and large calving front retreat, lead to an increase in ice discharge. The reconciled satellite observations and modeling results reveal that the continued basal melting in TIS drove long-term acceleration of ice discharge over the period from 1963 to 2015”

But the “modelling results” in the manuscript do not involve ice dynamics, so claims regarding causation cannot reasonably be made. Does reconciled refer to filling in the end of the new record with 1989 data or does it refer to considering the remote sensing and ocean modelling together?

My own thoughts about the circumstantial case may (or may not!) be helpful. What I see is a calving event between 1973 and 1989 that caused loss of ice shelf contact with a long section of near-front left margin. Because the stress balance is non-local, the ice flow response to this change in boundary conditions would be instantaneous and shelf-wide, and could explain the observed shelf-wide speedup (new data presented here). After that, margins contact is regained and ice speed declines (new and earlier data together). Near the front, ice speed then remains around the long-term average.

In the grounding zone, speed increases around the time of the calving event, then declines as elsewhere on the ice shelf (new data(but then picks back up again (new and earlier data together). An additional driver is required to explain the speedup in the grounding zone and basal melting may be it. This realisation is new and only possible because of the new data presented in this manuscript. The explanation that works near the front won’t work here but basal melting could. The bigger melt events ~1996 and ~2003 in Figure 2 may have caused those small pinning points near the grounding line, or driving stress at the grounding line, or shear margin resistive stresses to change. These are all are good suspects for driving change that matters to the ice flux across the grounding line but without an ice dynamics model or additional data analysis, we can’t affirm that basal melting is the driver and can’t make any claim about why (what process) it has had this effect.

Rather than shelf-wide, area-average basal melt rate, it might be more helpful to show the time series of basal melt rate in the grounding zones and margins, where melt really matters (Feldman et al., 2022; Haseloff and Sergienko, 2018). The leading EOF in Figure 3 of Gwyther et al. (2018) is quite interesting.

The shelf-wide and local patterns will be the same because these places dominate the melt signal but this would be the more ice-flux minded result to show, if possible.

Examining the maps of speed change in Figure 1, the area of pinning points near the grounding line appear to have the same sense of change as the fully floating ice so I don't think the response to calving between 1973 and 1989 did anything special here but the panels in the figure are small and I could be wrong.

Suggested improvements

The fundamental improvement required, on my view, is to include some glaciological reasoning in the manuscript. Some ideas are presented in the "Analytical approach" section of this review. More completely, Totten Glacier basin modelling could be undertaken to quantify system responses to various drivers of change, with the aim of uniquely linking causes and observed variations.

Clarity and Context

A few notes on wording that I found confusing

lines 24 to 27: "We found a persistent long-term ice discharge trend from 1963–2015 at an average rate of 68.1 ± 3.6 Gt/y and an acceleration of 0.16 ± 0.02 Gt/y making TG the greatest contributor to global sea level rise in EA. We attribute the long term acceleration to"

What is an "ice discharge trend"? Ice discharge is just the amount out through the gate. Is this a long term average rate?

line 55 to 58: "A longer satellite observation record is needed to determine if these serious changes occurred before 1989, implying that the Totten Glacier has responded to climate change in an accelerated way, similar to rapidly changing glaciers in the Amundsen Sea sector in WA, such as Pine Island and Thwaites"

This is a complicated sentence. I think the intent is to state that a longer record will put the recent variability in a better context. The glaciers should be given their full names. Pine Island and Thwaites Glaciers.

line 62: "Furthermore the input–output (IO) method..."

Further from what? The sentence does not quite fit with the rest of the paragraph. A short paragraph about I-O method would help here. The fact that input-output has been abbreviated suggests that there might have been such a paragraph in an earlier draft.

line 91: "high accuracy of 4–17 m/y" This is uncertainty rather than accuracy. The word accuracy should be replaced everywhere with the word uncertainty.

line 126, 128: Please add the % change for these numbers as well as the others. Are all % differences reported in the text relative to the long-term average as stated in line 162? It would be helpful to state this somewhere.

line 125: This is not ice flow dynamics as the word is generally used because there is no force budget analysis or simulation. This section is about changes in ice flow over time, which could be called kinematics.

line 130: "partially attributed to" Causation has not been demonstrated, so the statement should be that the speedup is in the same time range as large retreat of the calving front.

lines 137 and 138: "may have reduced the buttressing potential of TIS and cause" As discussed above, more thought and analysis is required.

line 166: "calving front retreat and advection" I think this should be calving front retreat and advance.

line 233 to 235: "we suggest that the observed mCDW on the continental shelf during our study period intruded into the cavity and caused the ice shelf basal melting modelled by COREv2 (Fig. 2D)"

Does "suggest" mean in the same manner as was modelled by Gwyther et al.? Those authors provided a detailed analysis.

line 284: "high ice dynamics" What is this?

line 287: Nothing has been established about ice shelf stability.

We appreciate the constructive comments and suggestions from referees. Our manuscript will be much improved by their input. We have made changes to our manuscript. In the following responses, we use “**bold**” text for comments, “non-bold” text for our responses, and “*italic*” for changed text in the manuscript.

REVIEWER COMMENTS

Reviewer #1 (Remarks to the Author):

Using satellite data, the authors estimated the ice velocity of Totten Glacier from the 1960s. The authors present estimated historical glaciological changes and past ocean observations. This kind of work investigating the history of glaciological and oceanographic changes is valuable. I recommend that this manuscript for published in Nature Communications after revision. However, I have a few major comments about their oceanographic analysis. As I am an oceanographer, I only comment on the ocean part of this work.

Major comment

Line 232: Authors state that mCDW was observed in front of the TIS. I downloaded the data from WOD but I could not find the same observations described in Fig S3. I found similar data in 110W (not 110E). I suspect that authors seem to flip the sign of longitude and they may be plotting the observations from other parts of the Southern Ocean. If my guess is correct, I recommend authors remove Fig. S3 and revise ocean related part of their text. Based on my understanding, I do not think no oceanographic observations exist in front of the TIS before the 1980s.

Response:

Thanks for the comment. We checked and there is no CTD data collected during our study period. We deleted the relevant text. We used Fig. S3 to depict the mCDW observations made in austral summer of 1965-1996 and new basal melting modelling results of separate ice shelf regions. Revised text for linkage between mCDW and basal melting is presented in the Discussion section (see response Line 224-237).

Minor comment

Line 228: Grounded icebergs?

Response:

Changed to “*grounded icebergs*”.

Line 224-237: If these are the correct datasets, it would be great if you could list the names of oceanographic campaigns and how they obtained these measurements. Are these ship-based CTD casts?

Response:

We checked and there is no CTD data collected during our study period. We revised this paragraph: “*Despite the spatial and temporal sparsity of oceanographic*

observations in our study area, we are able to use six conductivity, temperature, and depth (CTD) profiles (IDs 1–6) in an extended region (Fig. S3), mostly collected during austral summer of 1995-1996 (ref.⁴⁵), from the World Ocean Database (WOD) supported by the NOAA Climate and Global Change Program (<https://www.ncei.noaa.gov/>). Limited by fast ice and grounded icebergs, the CTD profiles were not deployed close to the shelf front. They reveal the presence of mCDW of relatively high temperature ($> \sim 0^{\circ}\text{C}$) and salinity ($> \sim 34.5$ PSU) at $\sim 300 - 600$ m depth, near the continental shelf break and in front of TIS (Figs. S3B and S3C). We further performed a regional COREv2 simulation from 1960 to 2007 (Fig. S3D) and show that the shelf-wide pattern (Fig. 2D) is dominantly attributed to the high melt signals in the grounding line region (~ 11 m/y) and margin regions (~ 5 m/y) (Fig. S3E) where ice velocity and ice-flux can be altered. The melt rate in the shelf front region is low at ~ 2 m/y. Given the same bathymetry of the continental shelf, and the entrances to and troughs underneath the ice shelf cavity, we suggest that during our study period warm mCDW water may have crossed the continental shelf and intruded into the cavity, resulting in increased basal melting and changing in driving stress at the grounding line or resistive stresses along shear margins⁵⁰⁻⁵⁴. Therefore, the mCDW induced basal melting process is responsible for the acceleration during our study period from 1963 to 1989 and further for the long-term acceleration trend of nearly 6 decades in the grounding line region of the Totten Glacier (Fig. 2C).”

Line 287: As mCDW-induced basal melting ...: See my comment above. Authors may need to remove all ocean-related discussions if observations are located in the wrong places.

Response:

We have added reasoning as how mCDW has invaded the cavity of TIS based on the CTD observations in 1996, entry to and trough underneath the ice shelf, and our newly added basal melt rates with separate regions of TIS (see above response). In that way we established the linkage between the mCDW and basal melting.

Reviewer #2 (Remarks to the Author):

Review of Li et al., Nature Communications

Satellite record reveals 1960s initiation of acceleration in Totten Glacier, Antarctica

General Comments

Li et al. utilise historic Landsat and ARGON optical imagery to track ice flow with a previously published hierarchical matching and network densification method. A novel contribution to the field is made as ice velocity of Totten Glacier, East Antarctica, is calculated between 1963 and 1989 for the first time, thereby significantly extending the time series of velocity observations available here.

Velocity measurements reveal an acceleration between 1963 and 1989 on Totten Ice Shelf both near the grounding line and near the ice front, but little change is observed on the grounded portion of Totten Glacier. In the context of the complete time series (extending to 2018), velocity peaked in 1989 near the ice front, but accelerated consistently near the grounding line, so long-term acceleration is only clear immediately seaward of the grounding line, and not at the shelf front. The authors attribute the long-term acceleration to the presence of warm Circumpolar Deep Water, and the associated shelf area loss and basal melt rate.

The manuscript would be improved through consideration of the following points:

1. The methodology is sound, although further transparency in error reporting is recommended in some instances (see specific comments on L60, 79 & 89) given unavoidable issues in the quality of historical imagery which gives rise to significant errors, particularly in the case of the ice velocity in 1989.

Response:

In addition to uncertainty description in the Method section and Table S1, we have added the following text in places as suggested.

L60: We revised the sentence: “..... *early optical images collected on films by the ARGON intelligence satellites during the 1960s (ref. ¹⁹) and the Landsat satellites in the 1970s and 1980s can be used to recover historical glacier topography and ice flow velocity fields^{9,20,21} with comparable uncertainties of earlier and recent velocity products^{1,22,23}.*”

L79 and L89: We added the following sentence at the end of the paragraph: “*Uncertainties of the produced velocity maps range from 4 m to 79 m (Table S1), which were estimated from the orthorectification error of the images, feature identification and matching errors, and timespan according to the error propagation law described in the Methods section.*”

2. More spatial nuance is required in the interpretation of ice dynamics in the extended velocity time series. Interpretation should clearly distinguish the spatial variation in long-term acceleration trends across Totten Glacier and its ice shelf i.e.

long-term acceleration occurs near the grounding line, but not near the shelf front. The lack of initial or long-term acceleration on the grounded ice should also be emphasised, given that discharge of grounded ice directly impacts Antarctica's contribution to sea-level rise, unlike the varying discharge of the floating portion of Totten Ice Shelf.

Response:

1) To clearly state the long-term acceleration near grounding line induced by basal melting and short speed up in ice shelf front caused by calving front retreat, we made following revisions.

In Abstract: based on our additional analysis and modelling results, we revised the text to *"We attribute the long-term acceleration near grounding line from 1963 to 2018 to basal melting induced by warm modified Circumpolar Deep Water, as evidenced by early shipboard CTD observations and basal melting modeling. The speed up in the shelf front region during 1973-1989 was caused by a large calving front retreat."*

In Results and Discussion sections: in response to this and other comments, we have presented our new results of spatial and temporal analysis as well as new modelling efforts in basal melting and ice shelf velocity simulation. Your suggested trend is now well stated in these two sections.

In the Conclusion section: *"..... We found an ice shelf-wide velocity increase trend during 1963–1989. The acceleration in ice shelf front for 1973-1989 is attributed to the large calving front retreat, while that in the grounding line region is mainly caused by the mCDW-induced basal melting, leading to an increase in ice discharge. The reconciled satellite observations and modeling results reveal that the continued basal melting in TIS drove the long-term acceleration of near grounding line and associated ice discharge over the period from 1963 to 2018."*

2) To emphasize "the lack of acceleration on the grounded ice", we made the following revisions.

L126: We added a phrase to emphasize the point: *"....., indicating insignificant variation of discharge from the upper part of grounded ice which would contribute to sea-level rise."*

L138: We deleted the buttressing part and revised the sentence to *"Loss of this "active shelf-ice" area (104 km² or ~2%) caused velocity responses throughout the ice shelf^{36,37}."*

L144: We revised this sentence to *"However, the velocity on grounded ice and near grounding line showed no significant changes, indicating no variations in discharge and direct contribution to sea-level rise from this part of the glacier."*

3. More clarity on the cause(s) of acceleration is required in the discussion. Good evidence, both in situ and modelled data, is provided to describe the behaviour of both ice shelf area and basal melt rates, but discussion of these explanations requires spatial nuance and further statistical confirmation (see specific comments on L174 & 183). Does shelf area have a statistical relationship with velocity observations near the grounding line (Box 2), as well as near the shelf front (Box 1)? Does the basal melt rate have a statistical relationship with velocity observations in both locations?

Response:

We added new analysis results to address the above comments

L174: We performed an additional correlation analysis between velocity in Box 2 (grounding line) and shelf area change: *“There is a long-term increasing trend of velocity over the period of 1963-2018 which is consistent with that of the modelled melt rate from 1960 to 2007 (ref. ³⁸), simulated with COREv2 forcing⁴² (Fig. 2D). The simulated melt rate shows lower basal melting of ~5 to 6 m/y, Thereafter melt rate estimated from multi-mission altimetry observations increased to 11.5 ± 2 m/y from 1994 to 2018 (ref. ⁴³) and 17.9 ± 1.2 m/y from 2005 to 2011 (ref. ¹⁷). We reconstructed a new velocity time series from that in Fig. 2C by subtracting its long-term linear trend (black dashed line) estimated by a regression, which then shows a timespan weighted correlation of $R^2=0.6$ with the shelf front area changes in Fig. 2B. Thus, the acceleration during 1973–1989 in Fig. 2C, with a velocity increase of 124 ± 8 m/y (~0%), appears to be, to a large extent, caused by the largest calving front retreat during the same period, whose area loss intruded into the “active” shelf ice region inside the PSI boundary and influenced ice kinematics in the grounding zone.”*

L183: Direct calculation of R^2 between velocity and basal melt rate resulted in $R^2 = 0.2$ in Box 1 and $R^2 = 0.01$ in Box 2. Although their linear trends appear similar, at least in Box 2, the low correlation may be caused by the large fluctuation in melt rate. This result is not presented in text.

In addition, we performed new basal melting modelling with separate regions of the ice shelf. The results are presented in the Discussion section: *“We further performed a regional COREv2 simulation from 1960 to 2007 (Fig. S3D) and show that the shelf-wide pattern (Fig. 2D) is dominantly attributed to the high melt signals in the grounding line region (~11 m/y) and margin regions (~5 m/y) (Fig. S3E) where ice velocity and ice-flux can be altered. The melt rate in the shelf front region is low at ~2 m/y.”*

Fig. S3. (A) Locations of six profiles of CTD observations collected in austral summer of 1995-1996 outside the TIS shelf front and on the continent shelf and slope, from the World Ocean Database (WOD). White box indicates the extend of the TIS region in (D). Also shown are the bed topography of grounded ice and the bathymetry underneath and outside the ice shelf, from BedMachine Antarctica³. (B) Potential temperature (°C) and (C) salinity in practical salinity units (PSU) of the six CTD profiles for the locations shown in (A). Profile labels are formatted as Profile ID-YearMonth. (D) Enlarged area of TIS with bathymetry and boundaries of regions for presenting basal melting modelling results in (E). (E) Modelled melt rates from 1960 to 2007 in the grounding line region, eastern and western margins, and shelf front region. The grounding line is from ref. ¹.

Reproducibility of the study is good due to the extensive material included in the Supplements. Reference to the literature is detailed and comprehensive throughout, which contextualises the findings nicely.

Response:

Thanks.

Specific Comments

L1: Results show velocity acceleration dating back to 1963, but do not prove acceleration began at this time. Acceleration prior to 1963 is feasible. Discussion focussed on acceleration observed on Totten Ice Shelf, rather than the grounded glacier. Suggest reword title to: ‘Satellite record reveals 1960s acceleration of Totten

Ice Shelf, East Antarctica'

Response:

As suggested, the title is changed to "*Satellite record reveals 1960s acceleration of Totten Ice Shelf, East Antarctica*"

L19-20: Reword for clarity. '... hinders the mass change forecasting and estimations of its contribution to global sea-level rise.'

Response:

The sentence is accordingly changed to "*There is a lack of knowledge of long-term mass balance in the region which hinders the mass change forecasting and estimations of its contribution to global sea-level rise.*"

L21: Is initiation of acceleration shown? Acceleration prior to 1963 isn't disproven in the study.

Response:

Agree. The sentence is now changed to "*Here we show that this acceleration trend in TG has occurred since the 1960s.*"

L28: Acceleration attributed shelf area loss and basal melt rates in text. Both should be mentioned explicitly here.

Response:

Based on our additional analysis and modelling results, we revised the text to "*We attribute the long-term acceleration near grounding line from 1963 to 2018 to basal melting induced by warm modified Circumpolar Deep Water, as evidenced by early shipboard CTD observations and basal melting modeling. The speed up in the shelf front region during 1973-1989 was caused by a large calving front retreat.*"

L34: The inclusion of references would be beneficial here.

Response:

"*Rignot et al., 2019¹*" is added.

L37: The uncertainty associated with this figure should be emphasised here.

Response:

It is rephrased to "*A reconciled solution shows a mass change rate with a large uncertainty, 5 ± 46 Gt/y, in East Antarctica (EA)⁴.*"

L40: Expand on the meaning of 'important' in this sentence. Reference(s) should also be included.

Response:

It is rephrased with an addition of reference: "*Furthermore, studies^{1,7} have noted that the Totten Glacier (TG), fed by the largest drainage basin in Wilkes Land⁸ ...*"

L45: How will understanding glaciological responses in the 1960s help forecast to future contributions to sea-level rise, when observations from the 1990s are already

available?

Response:

We deleted the “forecasting” part of the sentence.

L47: Is the runaway effect associated with MISI already in effect here given the long-term rate of acceleration is not constant according to Figure 2?

Response:

We discussed this runaway effect in the Discussion section. We suggest that MISI may have already been in effect in 1960s with the eastern lobe in grounding zone. Since this is a general statement which is based on references, we would like to suggest that the sentence stays here as is.

L48: ‘... intrusion of the relatively warm mCDW...’

Response:

“*relatively*” is added in the sentence.

L51: Repetition from L40?

Response:

The sentence is deleted.

L55-58: Sentence needs rewording for clarity. Why exactly are velocity observations prior to 1989 necessary? How does this relate to the behaviour at Pine Island and Thwaites?

Response:

We deleted “Pine Island and Thwaites Glaciers”. To also address Reviewer 3’s comment, the sentence is simplified to “..... *However, a longer satellite observation record is needed to put the recent variability in a better context.*”

L58: Reword to ‘recent satellite data’, rather than newer.

Response:

It is changed accordingly.

L60: Do the earlier optical films yield velocity data with comparable errors to modern techniques?

Response:

We revised the sentence: “..... *early optical images collected on films by the ARGON intelligence satellites during the 1960s (ref. ¹⁹) and the Landsat satellites in the 1970s and 1980s can be used to recover historical glacier topography and ice flow velocity fields^{9,20,21} with comparable uncertainties of earlier and recent velocity products^{1,22,23}.*”

L62: Sentence does not clearly relate to the rest of the paragraph.

Response:

We rerevised it to give this relationship: “..... *Furthermore, these ice velocity maps along with other supporting data will allow us to estimate mass balance during historical*

periods at both glacial and continental scales through the input–output (IO) method^{1,5}

L79: The creation of the overall velocity map (1963-1989) by averaging three periodic velocity maps should be stated explicitly here (in addition to in L96).

Response:

The sentence is revised: “*For the first time, a historical velocity map of the Totten Glacier region from 1963 to 1989 is generated by weight-averaging three periodic maps (Fig. 1A),*”

L79: Brief mention of error calculations is required in this paragraph, in addition to the information included in the Materials and Methods section and Table S1.

Response:

We added the following sentence at the end of the paragraph: “*Uncertainties of the produced velocity maps range from 4 m to 79 m (Table S1), which were estimated from the orthorectification error of the images, feature identification and matching errors, and timespan according to the error propagation law described in the Methods section.*”

Figure 1A: White text is difficult to read, and grounding line colour is difficult to distinguish from the colour depicting fast-flowing velocity.

Response:

We tried to change the white text to black. Then in some background it is also not clear. So, we have now fixed white and black colors depending on background. We changed the grounding line pattern by adding an outer boundary. It can now stand out from the fast-flowing velocity.

L86: A brief description of these processing techniques is required here.

Response:

We added following sentences: “*..... For instance, we adopted a semi-automatic algorithm for recognition and measurement of damaged fiducial marks on ARGON films; exterior orientation parameters were initially estimated from the ephemeris data and ground features, and then refined through a bundle adjustment procedure.*”

L89: What is the error associated with these velocity fields?

Response:

The errors for three maps are added: “*Uncertainties of the produced velocity maps range from 4 m to 79 m (Table S1), which were estimated from the orthorectification error of the images, feature identification and matching errors, and timespan according to the error propagation law described in the Methods section.*”

L91: How are these accuracy values calculated?

Response:

In answering the two related comments (L79 and L89) in the previous paragraph, we added the relevant text (see above response).

L98: It would be useful to mark this extended area in Figure 1.

Response:

We added a thin black dashed line in Fig. 1A to mark the boundary of this extended area. In text we revised the sentence: “*The extended area (outside thin black dashed line in Fig. 1A) is covered by using a regional velocity map*^{29,30}.”

Fig. 1. Reconstructed historical ice velocity record revealed an active ice dynamic state in the Totten Glacier region. (A) Ice flow velocity field from 1963 to 1989 with glacier and ice shelf centerlines marked AA' and BB', respectively, Box 1 in shelf front (~50 km from shelf front), Box 2 near grounding line (~4 km from grounding line) and shelf fronts showing the largest retreat during 1973–1989. (B) Velocity difference map (1973–1989 minus 1963–1973), illustrating an ice shelf-wide increase of 60 ± 11 m/y, due to the shelf front retreat. Red denotes acceleration and blue denotes deceleration. (C) Velocity difference map (1989 minus 1973–1989), showing that the ice velocity in 1989 remained high, with a difference of only -15 ± 55 m/y. (D)

Ice velocity changes from 1963 to 1989 along centerline BB' of the eastern tributary glacier. (E) Ice velocity changes from 1963 to 1989 along centerline AA' of the main trunk of Totten Glacier. The backgrounds of A–C are from the LIMA mosaic³². The ice shelf fronts in (A) are digitized from the ARGON and Landsat orthoimages. The grounding line is from ref.³³. The Passive Shelf Ice (PSI) boundary (green line) in (A) is from ref.³⁴. Bathymetry data on the continental shelf in (A) are from ref.³⁶.

L101: Should read: ‘If not corrected, the implication is that the overestimated historical velocity...’?

Response:

It is so corrected as suggested.

L126: The observation that velocity over grounded ice changed very little should be emphasised given that dynamic variation on grounded ice contributes directly to sea-level rise, unlike floating ice.

Response:

We added a phrase to emphasize the point: “....., *indicating insignificant dynamic variation on discharge from the upper part of grounded ice and direct contribution to sea-level rise.*”

L127: Figure 1B & D shows acceleration on the fast-flowing grounded glacier ice was also minimal.

Response:

We revised the sentence to focus on the ice shelf only: “*However, acceleration was mainly found on the ice shelf, with an average velocity increase of 60 ± 11 m/y (~7%) from 1963-1973 to 1973-1989 for the floating ice.*”

Figure 1B&C: Larger font required for title. Change titles to ‘Difference between 1963-1973 and 1973-1989 velocity’ and ‘Difference between 1973-1989 and 1989 velocity’. For clarity, note in legend that red denotes acceleration, while blue denotes deceleration.

Response:

Changes in titles of Figs. 1B and 1C are made: larger font and suggested text changes. In the caption of Fig. 1B we added “..... *Red denotes acceleration and blue denotes deceleration.*”

L136: Include PSI boundary on Figure 1A.

Response:

PSI boundary (green) was in the previous version, but not clear in the legend. Now we changed background color of legend and put PSI as the first in legend to make it more noticeable.

L138: Note that the loss of buttressing potential is unlikely to be the case because the velocity of the grounded ice was minimally affected.

Response:

Agreed. We deleted the buttressing part and revised the sentence to “*Loss of this “active shelf-ice” area (104 km² or ~2%) caused velocity responses throughout the ice shelf^{36,37}.*”

L140: Error too large to discuss this result with confidence.

Response:

We revised the text to remove the velocity change and large error: “*Ice velocity on the ice shelf remained high in 1989, at least 4 years after the retreat, without significant changes from the 1973–1989 map (Fig. 1C).*”

Figure 1E: Text in legend is partially cropped. How has the width of the grounding zone been determined?

Response:

We fixed the partially cropped text. We also changed the label to “Grounding line” in Figures 1D and 1E. In text we specifically described it as “*a 20 km zone upstream and downstream from the grounding line*”.

L142: Figure 1D should be referred to in the text prior to Figure 1E.

Response:

Considering the context of sentences, we may first finish the velocity description along the main trunk and on the ice shelf, and then that of the tributary glacier. To do it in this order, we swapped the order of “D” and “E” in Figure 1.

L144: Emphasis should be placed on this result given that changes to grounded ice directly contribute to sea level rise, and not changes in floating ice.

Response:

We revised this sentence to “*However, the velocity on grounded ice and near grounding line showed no significant changes, indicating no variations in discharge and direct contribution to sea-level rise from this part of the glacier.*”

Figure 2B: Change title to ‘Shelf area change’

Response:

It is so changed.

L165: Note that the acceleration is, therefore, not constant near the shelf front throughout the time period.

Response:

Thanks. We added this statement in the text.

L168: How are calving activities related to the presence of CDW, if at all?

Response:

We performed new basal melting modelling with separate regions of the ice shelf. The results are presented in the Discussion section: “*We further performed a regional COREv2 simulation from 1960 to 2007 (Fig. S3D) and show that the shelf-wide pattern*

(Fig. 2D) is dominantly attributed to the high melt signals in the grounding line region (~11 m/y) and margin regions (~5 m/y) (Fig. S3E) where ice velocity and ice-flux can be altered. The melt rate in the shelf front region is low at ~2 m/y.”

Since melt rate is not a significant factor in shelf front, we do not discuss it here in the “Ice flow kinematics” section.

L174: Replace ‘retread’ with ‘retreat’?

Response:

Thanks. Corrected.

L174: Can the R² between velocity increase and calving front retreat be calculated here?

Response:

Thanks for the suggestion. It was a bit more complicated than we thought to get the right result. It is explained in the text: “*There is a long-term increasing trend of velocity over the period of 1963-2018 which is consistent with that of the modelled melt rate from 1960 to 2007 (ref.³⁸), simulated with COREv2 forcing⁴² (Fig. 2D). The simulated melt rate shows lower basal melting of ~5 to 6 m/y. Thereafter melt rate estimated from multi-mission altimetry observations increased to 11.5±2 m/y from 1994 to 2018 (ref.⁴³) and 17.9±1.2 m/y from 2005 to 2011 (ref.¹⁷). We reconstructed a new velocity time series from that in Fig. 2C by subtracting its long-term linear trend (black dashed line) estimated by a regression, which then shows a timespan weighted correlation of R²=0.6 with the shelf front area changes in Fig. 2B. Thus, the acceleration during 1973–1989 in Fig. 2C, with a velocity increase of 124±8 m/y (~0%), appears to be, to a large extent, caused by the largest calving front retreat during the same period, whose area loss intruded into the “active” shelf ice region inside the PSI boundary and influenced ice kinematics in the grounding zone.*”

Fig. 2. Long-term ice velocity trend in TIS from 1963 to 2018: (A) ice velocity near the shelf front (Box 1 in Fig. 1A) during the study period of 1963–1989 shows acceleration caused by the largest calving front retreat and the highest velocity in nearly 6 decades; (B) shelf area change (loss or gain) due to calving front retreat (or advance) indicates correlation with the ice velocity trend near the shelf front in (A) ($R^2=0.8$); (C) ice velocity near grounding line (Box 2 in Fig. 1a) from 1963 to 2018 shows a long-term increasing trend (black dashed line) consistent with basal melt rate in (D), in addition to the acceleration induced by the calving front retreat during 1973–1989; (D) simulated area-averaged basal melt rate of TIS from 1960 to 2007 (ref. ³⁸) with an increasing trend (black dashed line).

Here we show the velocity time series in Box 2 after subtraction of the long-term trend.

L183: Can the R^2 between basal melt rates and velocity observations be calculated?

Response:

Direct calculation of R^2 between velocity and basal melt rate resulted in $R^2 = 0.2$ in Box 1 and $R^2 = 0.01$ in Box 2. Although their linear trends appear similar, at least in Box 2, the low correlation may be caused by the large fluctuation in melt rate. This result is not presented in text.

Line 188: A discharge increase is not certain due to overlapping error margins (Figure 3A) so this cannot be stated with certainty.

Response:

Agree. We changed it to “..... ice discharge in the Totten Glacier was at a high level, close to the reference SMB that is the long-term average SMB from 1979 to 2016 (Fig. 3A).”

L190: The fact that the surface mass balance is above the reference surface mass balance is not clear from Figure 3A. A zoom inset would perhaps be a useful addition.

Response:

Thanks for your suggestion. A zoom inset is added in Figure 3A.

Fig. 3. (A) Ice discharge with uncertainty (blue line and shaded margin) and SMB from RACMO2.3 p2 (red line and shaded margin) in the TG basin during 1963–1989.

Reference SMB (black dashed line) is averaged during 1979–2016. Average SMB (red dashed line) is calculated over the period of 1979–1989. Zoom inset “a” shows details for 1972–1974. (B) Cumulative Discharge (blue line), cumulative SMB (red line) and total mass change (net mass gain, black line) for the same period. Zoom inset “b” shows net mass gain with uncertainty (shaded margin) of the last two years.

Figure 3A: Include full y axis label

Response:

We made the full label for y axis: “Discharge and SMB (Gt/y)” in Figure 3A.

Figure 3B: The positive gradient of the total net mass (black line) is not clear. Again, a zoom inset or similar would perhaps be a useful addition here.

Response:

A zoom inset is added in Figure 3B.

Table 1/Figure 3: Figure captions need to be more informative throughout, stating what each of the coloured lines represent, for example, and not include any interpretation of results.

Response:

We revised both captions accordingly.

Table 1. Mass balance (MB) estimation from three velocity maps. F is ice flux across the flux gate (Fig. 1A). $dMFG$ is the correction applied to F . D is ice discharge across the grounding line ($D = F + dMFG$). MB is mass change rate ($MB = SMB - D$).

Fig. 3. (A) Ice discharge with uncertainty (blue line and shaded margin) and SMB from RACMO2.3 p2 (red line and shaded margin) in the TG basin during 1963–1989. Reference SMB (black dashed line) is averaged during 1979–2016. Average SMB (red dashed line) is calculated over the period of 1979–1989. Zoom inset “a” shows details for 1972–1974. (B) Cumulative Discharge (blue line), cumulative SMB (red line) and total mass change (net mass gain, black line) for the same period. Zoom inset “b” shows net mass gain with uncertainty (shaded margin) of the last two years.

How do Table 1 and Figure 3B align? The straight lines in Figure 3B are not representative of the varying SMB/Discharge rates displayed in Table 1.

Response:

We guess you mean the lines in Fig. 3B are “too straight”. This is because the vertical axis scale is set small because of the available page size. Here we stretch the vertical scale. The SMB (red) and MB (black) lines show some degree of variations, while discharge (blue) stays “straight”. Zoom inset (a) is taken in a transition year for discharge and it does not show a big difference. Zoom inset (b) is taken in years of varying SMB, so we can see changes. We hope this answers the question.

Line 234: Is there any evidence that the CDW reaches the grounding zone?

Response:

The is no direction observations of CDW in the cavity underneath the ice shelf. But it was found on continental shelf. There are other publications reporting CDW here based on their observations (on continental shelf) and modelling results. Our basal melting modelling results now show substantially larger melt rate in grounding zone than shelf front. We revised the text: “*Despite the spatial and temporal sparsity of oceanographic observations in our study area, we are able to use six conductivity, temperature, and depth (CTD) profiles (IDs 1–6) in an extended region (Fig. S3), mostly collected during austral summer 1996 (ref.⁴⁵), from the World Ocean Database (WOD) supported by the NOAA Climate and Global Change Program (<https://www.ncei.noaa.gov/>). Limited by fast ice and grounded icebergs, the CTD profiles were not deployed close to the shelf front. They reveal the presence of mCDW of high temperature ($> \sim 2^{\circ}\text{C}$) and salinity ($> \sim 34.5$ PSU) below $\sim 300 - 500$ m near the continental shelf break and in front of TIS (Figs. S3B and S3C). We further performed a regional COREv2 simulation from 1960 to 2007 (Fig. S3D) and show that the shelf-wide pattern (Fig. 2D) is dominantly attributed to the high*

melt signals in the grounding line region (~ 11 m/y) and margin regions (~ 5 m/y) (Fig. S3E) where ice velocity and ice-flux can be altered. The melt rate in the shelf front region is low at ~ 2 m/y. Given the same bathymetry of the continental shelf, and the entrances to and troughs underneath the ice shelf cavity, we suggest that during our study period warm mCDW water may have crossed the continental shelf and intruded into the cavity, resulting in increased basal melting and changing in driving stress at the grounding line or resistive stresses along shear margins⁵⁰⁻⁵⁴

Fig. S3. (A) Locations of six profiles of CTD observations collected in austral summer of 1995-1996 outside the TIS shelf front and on the continent shelf and slope, from the World Ocean Database (WOD). White box indicates the extend of the TIS region in (D). Also shown are the bed topography of grounded ice and the bathymetry underneath and outside the ice shelf, from BedMachine Antarctica³. (B) Potential temperature ($^{\circ}\text{C}$) and (C) salinity in practical salinity units (PSU) of the six CTD profiles for the locations shown in (A). Profile labels are formatted as Profile ID-YearMonth. (D) Enlarged area of TIS with bathymetry and boundaries of regions for presenting basal melting modelling results in (E). (E) Modelled melt rates from 1960 to 2007 in the grounding line region, eastern and western margins, and shelf front region. The grounding line is from ref. ¹.

L236: Are you able to separate the roles of basal melt vs. front retreat? Do they have different influences at different locations on Totten Glacier and its ice shelf?

Response:

1) Using velocity observations, we are able to say that calving front retreat caused acceleration in both shelf front and grounding zone (see above responses and text in “Ice flow kinematics” (Fig. 2)

2) Our new basal melting modelling efforts resulted in the conclusion that the long-term melting is responsible for acceleration in grounding zone, but it has much less impact on shelf front (Fig. S3).

3) Our new ice shelf modelling results show that calving front retreat caused acceleration in shelf front, but the speed up in grounding zone is small (<10 m/y).

Therefore, we summarize in abstract and conclusions that the long-term (1963-2018) acceleration in grounding zone is mainly induced by basal melting. The speed up in shelf front from 1973-1989 is caused by the large front retreat. In Results and Discussion sections we also presented the results of the lower impact of ice front calving on grounding zone (observations) and modelled melt rate of 2 m/y in the shelf front region (modelled).

L259: Is there evidence or a reference for grounding line retreat here?

Response:

Grounding line retreat detected by InSAR technique is documented in Li et al. (2015)². We added it in text and reference.

Li, X., Rignot, E., Morlighem, M., Mouginot, J. & Scheuchl, B. Grounding line retreat of Totten Glacier, East Antarctica, 1996 to 2013. *Geophys. Res. Lett.* 42, 8049–8056 (2015).

L260: Expand upon what is occurring at Pine Island Glacier, and how the evidence at Totten Glacier aligns.

Response:

We revised the text to make this alignment: “..... *We further suggest that the TG is currently experiencing a process that occurred earlier at Pine Island Glacier, another marine-based and rapidly changing glacier on the AIS⁵⁶⁻⁵⁸ where basal melting induced grounding line retreat and speedup occurred for several decades prior to sustained large scale ice shelf retreat was observed since 2017⁵⁸. Analogously, since 1990, the observed TIS calving front positions have been changing modestly, mostly between the shelf fronts of 1973 and 1989 (Fig. S5), and likely with influence from sea ice dynamics¹⁴. Meanwhile the mCDW-induced basal melting caused grounding line retreat and acceleration in the TIS for over three decades. Therefore, more rapid calving activities may be expected as basal melting persistently weakens the stability of the ice shelf, resulting in an imbalanced shelf front retreat. Intensified monitoring of the TIS is recommended in the next decades.*”

L282: Spatial nuance required here. Long-term acceleration only observed in Box 2 (near grounding line), not Box 1 (near shelf front).

Response:

Agree. We revised the sentence: “*We found an ice shelf-wide velocity increase trend during 1963–1989. The acceleration in ice shelf front for 1973-1989 is attributed to the*

large calving front retreat, while that in the grounding line region is mainly caused by the mCDW-induced basal melting, leading to an increase in ice discharge. The reconciled satellite observations and modeling results reveal that the continued basal melting in TIS drove the long-term acceleration near grounding line and associated ice discharge over the period from 1963 to 2018.”

Reviewer #3 (Remarks to the Author):

Key Results

The manuscript reports on new insights developed using velocity from historical image materials that extend the observational record of Totten Glacier flow much earlier than previously available.

On my view, the data support the following claims

- 1) There is an overall trend toward negative net mass balance that began earlier than previously recognised.**
- 2) A shelf-wide speed up was coincident with calving that would have reduced drag provided by the left margin (Figure 1, Figure 2, Figure S5).**
- 3) While the shelf front speed reduced toward the long-term average after contact was regained with the left front margin (Figure 2, Figure S5), the grounding zone appears to have experienced a more persistent speed up despite that, requiring an additional explanation.**

Response:

We have incorporated these three points into Abstract and Conclusions. The speed up caused by basal melting in grounding zone is now supported by new melt rate modelling results specifically simulated with separate regions.

Based on the modelling results we revised the text in Discussion: “..... *We further performed a regional COREv2 simulation from 1960 to 2007 (Fig. S3D) and show that the shelf-wide pattern (Fig. 2D) is dominantly attributed to the high melt signals in the grounding line region (~11 m/y) and margin regions (~5 m/y) (Fig. S3E) where ice velocity and ice-flux can be altered. The melt rate in the shelf front region is low at ~2 m/y. Therefore, the mCDW induced basal melting process is responsible for the acceleration during our study period from 1963 to 1989 and further for the long-term acceleration trend of nearly 6 decades in the grounding line region of the Totten Glacier (Fig. 2C).*”

Fig. S3. (A) Locations of six profiles of CTD observations collected in summer 1996 outside the TIS shelf front and on the continent shelf and slope, from the World Ocean Database (WOD). White box indicates the extend of Totten Glacier in (D). Also shown are the bed topography of grounded ice and the bathymetry underneath and outside the ice shelf, from BedMachine Antarctica³. (B) Potential temperature ($^{\circ}\text{C}$) and (C) salinity in practical salinity units (PSU) of the six CTD profiles for the locations shown in (A). Profile labels are formatted as Profile ID-YearMonth. (D) Enlarged area of Totten Glacier with bathymetry and boundaries of regions for presenting basal melting modelling results in (E). (E) Modelled melt rates from 1960 to 2007 in the grounding line regions, eastern and western margins, and shelf front region. The grounding line (red) is from ref. ¹.

This leads to new insight. Conclusions in the literature (shown in Figure 2), that velocity has a multi-year cycle of speed up and slow down associated with intrinsic variability, are limited by the short time span they cover. The new data presented here shows us that the higher frequency variability is superimposed on other time scales if variation, some of which appears to be a direct result of calving and some of which requires further explanation.

Response:

The long-term acceleration trend in grounding zone (and shear margins) is very clear, as evidenced by both observations in Box 2 and melt rate modelling (see Fig. 3 above).

Velocity variations exist in the shelf front region, amplified by the big calving front retreat (Fig. 2A and following new ice shelf modelling result).

We present the ice shelf modelling results in the Discussion section: “*Our ice shelf modeling results (Method and Fig. S6) reveal that the shelf front retreat during 1973-1985, which caused loss of ice shelf contact with a long near-front section of western margin, induced significant speed up (>300m/a) of shelf flow in proximity to the western margin (Fig. S6F). This instantaneous response explains the observed acceleration detected in Box 1 using the velocity maps (Fig. 2A). After that, margin contact is regained; ice speed declines and then remains around the long-term average. Furthermore, because of the non-local stress balance, the calving front retreat impacted the entire ice shelf and immediate grounded areas of grounding line with an average velocity increase of ~84 m/y (Fig. 6F). This explains the shelf-wide acceleration from 1963-1973 to 1973-1989 (Fig. 1B). The modelled velocity response in the grounding line region to the calving front retreat is less than 10m/y, lower than the observed increase Box 2 (Fig. 2C).*”

Fig. S6. (A) Model mesh for Totten glacier drainage basin and ice shelf using the 1973 ice shelf front. Mesh sizes range from 600 to 35000 m in the grounded area and 600 to 700 m in the ice shelf or over ice rises (see inset). (B) and (C) Spatial distribution of the ice rigidity parameter B and ISSM friction coefficient C , respectively, in the Totten ice shelf region. The grounding line is from ref¹ and rumples extents are simplified from ref. ⁷.

The ice surface and bed elevations are interpolated from Bedmachine Antarctica V3⁸. The velocity field used for inversion is the 1963-73 velocity map (Fig. S1A). The outside portion on grounded ice is filled with velocity of MEaSURES V2⁹. (D) Modelled velocity field of 1973 using 1973 shelf front. (E) Modelled velocity field of 1989 using 1989 shelf front with the large calving front retreat in the western margin. (F) Difference between the modelled velocity maps of 1989 (E) and 1973 (D). Ice shelf extent not covered by the 1963-73 velocity map (Fig. S1A) is not included in (D) to (F).

The recent velocity data (Figure 2) exhibits variability with time scales similar to those in basal melting predicted by an ocean cavity circulation model, making ocean heat in the ice shelf cavity a likely driver for change that is unlikely to be explained by changes near the shelf front.

Response:

The melt rate modelling result does indicate that basal melting has a very low impact on the shelf front region (Fig. S3E and see above response).

The Totten Glacier basin mass balance calculations produce another meaningful result, implying a longer term trend toward negative mass balance than previously concluded. Speculation about what may come next is overstated relative to the data presented here.

Response:

We deleted the overstatement and made the following relevant changes.

In Abstract: *“As the current acceleration and mass loss trend continues, intensified monitoring of ice-air-water interactions in the TG region is recommended in the next decades. ~~The sustained mass loss in the marine based Totten Glacier may lead to more rapid changes with marine ice sheet instability signatures and increased contribution to global sea level rise, as occurred in the Amundsen Sea sector in West Antarctica.~~”*

In Conclusions: *“We suggest that recently reported ice flow acceleration and mass loss trend in the TG basin and the Wilkes Land sector of East Antarctica since the 1980s may have started in the 1960s. As this trend continues, intensified monitoring of ice-air-water interactions in the TIS region is recommended in the next decades. ~~As mCDW induced basal melting continually causes ice shelf thinning and as the MISI process reduces the stability of the ice shelves in this sector, larger and damaging calving activities may occur more frequently. This strengthened effect of compound basal melting and calving may lead to more significant mass loss and a greater contribution to global sea level rise compared to those of the Antarctic Peninsula and the Amundsen Sea sector in West Antarctica.~~”*

Validity

The paper presents glacier kinematics associated circumstances. Contrary to several statements in the text, the “state of ice dynamics of the TG region” has not been established. Possible causes for the observed change have not been evaluated quantitatively (that is, using models or mechanical analysis of the observations). The circumstantial case (involving the new observations and published simulations of ice

shelf cavity circulation and basal melting) is not made as completely as it could be due to missing glaciological reasoning.

Response:

We performed your suggested modeling and analysis work. We added Martin Forbes as coauthor who performed ice shelf modelling. Overall, the following major changes are made in this revision to link the observed acceleration and discharge to the causes of basal melting in grounding zone and calving front retreat.

- 1) We performed additional spatial and temporal glaciological reasoning, including removal of basal melting trend, and then calculating correlation between new time series of velocity and area loss.
- 2) We further modelled the basal melting of TIS with separate regions of shelf front, grounding line and left and right margins. The results show high level melting concentrated in grounding zone and margins that match with the observed speed up in Box 2 and discharge estimates.
- 3) We added mechanical modelling of TIS, which is capable of reconstructing the velocity field of 1963-1973, and then shows that the calving front retreat during 1973-1985 did cause acceleration in the shelf front region and on the entire ice shelf.

Significance

The new observations presented here meaningful because they put recent change into a more climatologically relevant context, and because they offer more evidence with which to evaluate possible causes for the observed change. The longer time series challenges some recent focus on intrinsic variability only, showing that when the longer view is available, that variability may in fact be superimposed on a trend.

Response:

We performed additional analysis to prove this new long-term trend. The text in Results is revised. *“Conversely, the velocity near grounding line (Box 2 in Fig. 1A) was initially low during 1963–1973, 123 ± 21 m/y (~15%) below the long-term average velocity (Fig. 2C). There is a long-term increasing trend of velocity over the period of 1963-2018 which is consistent with that of the modelled melt rate from 1960 to 2007 (ref. ³⁸), simulated with COREv2 forcing⁴² (Fig. 2D). The simulated melt rate shows lower basal melting of ~5 to 6 m/y. Thereafter melt rate estimated from multi-mission altimetry observations increased to 11.5 ± 2 m/y from 1994 to 2018 (ref. ⁴³) and 17.9 ± 1.2 m/y from 2005 to 2011 (ref. ¹⁷). We reconstructed a new velocity time series from that in Fig. 2C by subtracting its long-term linear trend (black dashed line) estimated by a regression, which then shows a timespan weighted correlation of $R^2=0.6$ with the shelf front area changes in Fig. 2B. Thus, the acceleration during 1973–1989 in Fig. 2C, with a velocity increase of 124 ± 8 m/y (~0%), appears to be, to a large extent, caused by the largest calving front retreat during the same period, whose area loss intruded into the “active” shelf ice region inside the PSI boundary and influenced ice kinematics in the grounding zone. The ice velocity declines as elsewhere on the ice shelf in 1989, but then picks back up again. This recent acceleration after 1989, also reported in refs.^{1,5,6,16}, was induced by ice shelf basal melting caused by intrusion of mCDW from the continental shelf into an ice shelf cavity through an underwater trough^{7,8,44}. The combined long-term melt rates from 1960 to 2018*

based on modelling and satellite observations suggest that ice shelf basal melting may have existed in TIS as early as 1960. Here we demonstrate that the long-term acceleration trend in the grounding region induced mainly by ice shelf basal melting (Fig. 2C) started in 1963, 26 years earlier than reported."

Fig. 2. Long-term ice velocity trend in TIS from 1963 to 2018: (A) ice velocity near the shelf front (Box 1 in Fig. 1A) during the study period of 1963–1989 shows acceleration caused by the largest calving front retreat and the highest velocity in nearly 6 decades; (B) shelf area change (loss or gain) due to calving front retreat or advance indicates correlation with the ice velocity trend near the shelf front in (A) ($R^2=0.8$); (C) ice velocity near grounding line (Box 2 in Fig. 1A) from 1963 to 2018 shows a long-term increasing trend (black dashed line) consistent with basal melt rate in (D), in addition to the acceleration induced by the calving front retreat during

1973–1989; (D) simulated area-averaged basal melt rate of TIS from 1960 to 2007 (ref. ³⁸) with an increasing trend (black dashed line).

Data and methodology

The methods applied here are established and reviewed in the literature.

The “study period” is the time interval of the new data (1963 to 1989) but other ranges are used in some figures. This makes sense for Figures 1 and 2. But why does Figure S4 end in 2015 when the data in Figure 2 extend to 2018?

Response:

We extended the span of ice discharge calculation to 2018. Accordingly, Figure S4 was revised as follows.

“**Fig. S4.** (A) Long-term (1963–2018) SMB (RACMO 2.3 p2) and ice discharge derived from the 1963–1989 maps from this study and 1989–2018 maps from refs. ⁴⁻⁶. (B) Cumulative SMB, discharge, and mass balance are computed with the transition point of 1989 as the starting time for forward and backward cumulative mass change integration of each item minus reference SMB. The MB in the TG basin from 1963 to 2018 is dominated by long-term accelerated ice discharge, modulated by the SMB.”

Figure 4S is a bit misleading (unintentionally so, I think). As plotted, the early part of the record looks like a steady decline toward negative MB but the early data are average representations of intervals that must surely have experienced variability of the sort shown in the more recent part of the record.

Response:

The early part covers 1963-1989. We do have three different discharge rates of 1963-1973, 1973-1989 and 1989 that are derived from three different velocity maps (Table 1 and Fig. S4A). The rates are increasing, although the cumulative discharge during this period appears “steady” because of the extended time scale from 1963-2018. These three estimates of 27 historical years may not look “many” and cannot show annual variations, compared to estimates of one to several years timespan in recent 29 years, but they are

insightful to show us unprecedentedly the overall increasing trend in that period and a clear persistent long-term trend of six decades.

We added the following sentence to address your comment in Discussion: *“Although we cannot show short timespan (e.g., annual) variations, the overall discharge increase trend during the period is clearly demonstrated.”*

Analytical approach

Claims regarding causation are not as well supported as they should be (or could be). In the Conclusion, “We found an ice velocity increase trend during 1963–1989, which is attributed to the combined effect of the mCDW-induced basal melting and large calving front retreat, lead to an increase in ice discharge. The reconciled satellite observations and modeling results reveal that the continued basal melting in TIS drove long-term acceleration of ice discharge over the period from 1963 to 2015”

Response:

Claims regarding causation are now better supported by additional analysis, extended basal melting modelling and new ice shelf modelling (see above responses). We also revised the sentences in Conclusions: *“We found an ice shelf-wide velocity increase trend during 1963–1989. The acceleration in ice shelf front for 1973–1989 is attributed to the large calving front retreat, while that in the grounding line region is mainly caused by the mCDW-induced basal melting, leading to an increase in ice discharge. The reconciled satellite observations and modeling results reveal that the continued basal melting in TIS drove the long-term acceleration near grounding line and associated ice discharge over the period from 1963 to 2018.”*

But the “modelling results” in the manuscript do not involve ice dynamics, so claims regarding causation cannot reasonably be made. Does reconciled refer to filling in the end of the new record with 1989 data or does it refer to considering the remote sensing and ocean modelling together?

Response:

With the new ice shelf modelling results and extended basal melting simulations, our claims in Conclusions should be now supported by both observations and modelling: *“The reconciled satellite observations and modeling results.....”*

My own thoughts about the circumstantial case may (or may not!) be helpful. What I see is a calving event between 1973 and 1989 that caused loss of ice shelf contact with a long section of near-front left margin. Because the stress balance is non-local, the ice flow response to this change in boundary conditions would be instantaneous and shelf-wide, and could explain the observed shelf-wide speedup (new data presented here). After that, margins contact is regained and ice speed declines (new and earlier data together). Near the front, ice speed then remains around the long-term average.

Response:

Thanks for offering your thoughts. We incorporated this into our text for presenting the ice shelf modelling results in Discussion: *“Our ice shelf modeling results (Method and*

Fig. S6) reveal that the shelf front retreat during 1973-1985, which caused loss of ice shelf contact with a long near-front section of western margin, induced significant speed up (>300m/a) of shelf flow in proximity to the western margin (Fig. S6F). This instantaneous response explains the observed acceleration detected in Box 1 using the velocity maps (Fig. 2A). After that, margin contact is regained; ice speed declines and then remains around the long-term average. Furthermore, because of the non-local stress balance, the calving front retreat impacted the entire ice shelf and immediate grounded areas of grounding line with an average velocity increase of ~84 m/y (Fig. 6F). This explains the shelf-wide acceleration from 1963-1973 to 1973-1989 (Fig. 1B). The modelled velocity response in the grounding line region to the calving front retreat is less than 10m/y, lower than the observed increase Box 2 (Fig. 2C)."

In the grounding zone, speed increases around the time of the calving event, then declines as elsewhere on the ice shelf (new data(but then picks back up again (new and earlier data together). An additional driver is required to explain the speedup in the grounding zone and basal melting may be it. This realisation is new and only possible because of the new data presented in this manuscript. The explanation that works near the front won't work here but basal melting could. The bigger melt events ~1996 and ~2003 in Figure 2 may have caused those small pinning points near the grounding line, or driving stress at the grounding line, or shear margin resistive stresses to change. These are all are good suspects for driving change that matters to the ice flux across the grounding line but without an ice dynamics model or additional data analysis, we can't affirm that basal melting is the driver and can't make any claim about why (what process) it has had this effect.

Response:

Thanks again. The mCDW induced basal melting should have played a big role in grounding zone, as evidenced by the new melt rate modelling results. We revised the relevant text in Discussion: "... We further performed a regional COREv2 simulation from 1960 to 2007 (Fig. S3D) and show that the shelf-wide pattern (Fig. 2D) is dominantly attributed to the high melt signals in the grounding line region (~11 m/y) and margin regions (~5 m/y) (Fig. S3E) where ice velocity and ice-flux can be altered. The melt rate in the shelf front region is low at ~2 m/y. Therefore, the mCDW induced basal melting process is responsible for the acceleration during our study period from 1963 to 1989 and further for the long-term acceleration trend of nearly 6 decades in the grounding line region of the Totten Glacier (Fig. 2C)."

Regarding pinning points, there are rumples and pinning points (numbered from 1 to 5) located in the middle and front regions of the ice shelf (see following figure).

Pinning points

We have examined velocity on and around the rumples and pinning points from 1963 to 2015. There are no obvious changes (ungrounding and speed up) before and after “the bigger melt events between ~1996 and ~2003 in Figure 2”. The changes we see are the velocity decrease from the calving-induced acceleration to the long-term average velocity similar to what occurred in Box 1 (Fig. 2A).

Rather than shelf-wide, area-average basal melt rate, it might be more helpful to show the time series of basal melt rate in the grounding zones and margins, where melt really matters (Feldman et al., 2022; Haseloff and Sergienko, 2018)^{3,4}. The leading EOF in Figure 3 of Gwyther et al. (2018)⁵ is quite interesting. The shelf-wide and local patterns will be the same because these places dominate the melt signal but this would be the more ice-flux minded result to show, if possible.

Response:

Thanks for the suggestion. It is done and shown in above responses.

Examining the maps of speed change in Figure 1, the area of pinning points near the grounding line appear to have the same sense of change as the fully floating ice so I don't think the response to calving between 1973 and 1989 did anything special here but the panels in the figure are small and I could be wrong.

Response:

The velocity of our maps and recent maps are displayed in enlarged rumple and pinning points areas (see above responses). We haven't seen anything special.

Suggested improvements

The fundamental improvement required, on my view, is to include some glaciological reasoning in the manuscript. Some ideas are presented in the "Analytical approach" section of this review. More completely, Totten Glacier basin modelling could be undertaken to quantify system responses to various drivers of change, with the aim of uniquely linking causes and observed variations.

Response:

We appreciate your suggestion. They are very constructive and we did what you have suggested (see above responses and the revised manuscript).

Clarity and Context

A few notes on wording that I found confusing

lines 24 to 27: “We found a persistent long-term ice discharge trend from 1963–2015 at an average rate of 68.1 ± 3.6 Gt/y and an acceleration of 0.16 ± 0.02 Gt/y making TG the greatest contributor to global sea level rise in EA. We attribute the long term acceleration to” What is an “ice discharge trend”? Ice discharge is just the amount out through the gate. Is this a long term average rate?

Response:

We changed it to “*We found a persistent long-term ice discharge rate of 68 ± 1 Gt/y and an acceleration of 0.17 ± 0.02 Gt/y² from 1963–2018, making TG the greatest contributor to global sea level rise in EA.*”

line 55 to 58: “A longer satellite observation record is needed to determine if these serious changes occurred before 1989, implying that the Totten Glacier has responded to climate change in an accelerated way, similar to rapidly changing glaciers in the Amundsen Sea sector in WA, such as Pine Island and Thwaites”

This is a complicated sentence. I think the intent is to state that a longer record will put the recent variability in a better context. The glaciers should be given their full names. Pine Island and Thwaites Glaciers.

Response:

To also address Reviewer 2’s comment, we deleted “Pine Island and Thwaites Glaciers”. The sentence is simplified to “..... *However, a longer satellite observation record is needed to put the recent variability in a better context.*”

line 62: “Furthermore the input–output (IO) method...”

Further from what? The sentence does not quite fit with the rest of the paragraph. A short paragraph about I-O method would help here. The fact that input-output has been abbreviated suggests that there might have been such a paragraph in an earlier draft.

Response:

The I-O method is a widely applied method. There is also no space to describe it in detail in this paper. So, in addition to giving references, we revised the sentence to both make the context and explain the method: “..... *Furthermore, these ice velocity maps along with other supporting data allow us to estimate mass balance during historical periods at both glacial and continental scales through the input–output (IO) method^{1,5}.*”

line 91: “high accuracy of 4–17 m/y” This is uncertainty rather than accuracy. The word accuracy should be replaced everywhere with the word uncertainty.

Response:

It is replaced here and in another place.

line 126, 128: Please add the % change for these numbers as well as the others. Are all % differences reported in the text relative to the long-term average as stated in line 162? It would be helpful to state this somewhere.

Response:

L126: We added the % change for all these numbers: “*The velocity of grounded ice in the slow-flowing area (< 80 m/y, Fig. 1A) of the Totten Glacier changed little, on average 6 ± 16 m/y (~0%), from 1963 to 1989 as indicated by the two velocity difference maps (Figs. 1B and 1C)*”.

In the paragraph of long-term velocity analysis in Boxes 1 and 2 we made an overall statement: “*We calculate changes in percentage during our study period relative to the long-term average from 1963 to 2018.*”

line 125: This is not ice flow dynamics as the word is generally used because there is no force budget analysis or simulation. This section is about changes in ice flow over time, which could be called kinematics.

Response:

It is changed to “*Ice flow kinematics*”.

line 130: “partially attributed to” Causation has not been demonstrated, so the statement should be that the speedup is in the same time range as large retreat of the calving front.

Response:

We changed the sentence accordingly: “*This shelf-wide acceleration was in the same time range as a large calving front retreat between 1973 and 1989 (Fig. 1A), with an area loss of ~ 645 km² (12%).*”

line 166: “calving front retreat and advection” I think this should be calving front retreat and advance.

Response:

“*calving front retreat and advection*” is revised to “*calving front retreat and advance*”.

line 233 to 235: “we suggest that the observed mCDW on the continental shelf during our study period intruded into the cavity and caused the ice shelf basal melting modelled by COREv2 (Fig. 2D)”

Does “suggest” mean in the same manner as was modelled by Gwyther et al.? Those authors provided a detailed analysis.

Response:

We have performed an extended basal melting modelling with different regions. Based on the modelling results we revised the text: “*..... We further performed a regional COREv2 simulation from 1960 to 2007 (Fig. S3D) and show that the shelf-wide pattern (Fig. 2D) is dominantly attributed to the high melt signals in the grounding line region (~ 11 m/y) and margin regions (~ 5 m/y) (Fig. S3E) where ice velocity and ice-flux can be altered. The melt rate in the shelf front region is low at ~ 2 m/y. Therefore, the mCDW induced basal melting process is responsible for the acceleration during our study period from 1963 to 1989 and further for the long-term acceleration trend of nearly 6 decades in the grounding line region of the Totten Glacier (Fig. 2C).*”

line 284: “high ice dynamics” What is this?

Response:

It is changed to “ice flow acceleration”.

line 287: Nothing has been established about ice shelf stability.

Response:

Agree. We revised text and deleted the statement: “*We suggest that recently reported ice flow acceleration and mass loss trend in the TG basin and the Wilkes Land sector of East Antarctica since the 1980s may have started in the 1960s. As this trend continues, intensified monitoring of ice-air-water interactions in the TIS region is recommended in the next decades. As mCDW-induced basal melting continually causes ice shelf thinning and as the MISI process reduces the stability of the ice shelves in this sector, larger and damaging calving activities may occur more frequently. This strengthened effect of compound basal melting and calving may lead to more significant mass loss and a greater contribution to global sea level rise compared to those of the Antarctic Peninsula and the Amundsen Sea sector in West Antarctica.*”

Reference:

1. Rignot, E. *et al.* Four decades of Antarctic Ice Sheet mass balance from 1979–2017. *Proc. Natl. Acad. Sci.* **116**, 1095–1103 (2019).
2. Li, X., Rignot, E., Morlighem, M., Mouginot, J. & Scheuchl, B. Grounding line retreat of Totten Glacier, East Antarctica, 1996 to 2013. *Geophys. Res. Lett.* **42**, 8049–8056 (2015).
3. Feldmann, J., Reese, R., Winkelmann, R. & Levermann, A. Shear-margin melting causes stronger transient ice discharge than ice-stream melting in idealized simulations. *The Cryosphere* **16**, 1927–1940 (2022).
4. Haseloff, M. & Sergienko, O. V. The effect of buttressing on grounding line dynamics. *J. Glaciol.* **64**, 417–431 (2018).
5. Gwyther, D. E., O’Kane, T. J., Galton-Fenzi, B. K., Monselesan, D. P. & Greenbaum, J. S. Intrinsic processes drive variability in basal melting of the Totten Glacier Ice Shelf. *Nat. Commun.* **9**, 3141 (2018).

REVIEWERS' COMMENTS

Reviewer #1 (Remarks to the Author):

Using satellite data, the authors estimated the ice velocity of Totten Glacier from the 1960s. The authors present estimated historical glaciological changes and past ocean observations. This kind of work investigating the history of glaciological and oceanographic changes is valuable. I recommend that this manuscript for published in Nature Communications after minor revision. As I am an oceanographer, I only comment on the ocean part of this work.

Minor comment

- "Therefore, the mCDW-induced basal melting process is responsible for the acceleration during our study period from 1963 to 1989 and further for the long-term acceleration trend of nearly 6 decades in the grounding line region of the Totten Glacier (Fig. 2C)."

As there is no data between 1963-1989, I do not think authors can make a statement that mCDW inducted basal melting "is" responsible. I think they should say "is likely" responsible. I think this sentence is too strong.

Reviewer #2 (Remarks to the Author):

I have read through the updated manuscript, and the responses to my own comments on the initial draft of the manuscript. I am satisfied that my comments have been addressed sufficiently, particularly with regards to clarifying the spatial variation in the role of ocean melting vs ice shelf area, on ice velocity. The manuscript has also been significantly improved by the updates made to Figure 2, and the accompanying additional analysis.

From a scientific standpoint, I can now recommend publication once the few minor comments below have been addressed:

L36 – Should read 'The Antarctic Ice Sheet'.

L186 – full stop before thereafter.

L252 – 'primarily' is perhaps a better word than 'dominantly'.

L315 – note that while Box 2 is in the grounding line region, it is still located on floating ice, which means any velocity variability will not directly increase ice discharge. This sentence should be rephrased to highlight that this process can indirectly lead to increased ice discharge. Change throughout as necessary.

Note: line numbers refer to the non-tracked changes version of the updated manuscript.

We appreciate the constructive comments and suggestions from referees. Our manuscript will be much improved by their input. We have made changes to our manuscript. In the following responses, we use “**bold**” text for comments, “non-bold” text for our responses, and “*italic*” for changed text in the manuscript.

REVIEWER COMMENTS

Reviewer #1 (Remarks to the Author):

Using satellite data, the authors estimated the ice velocity of Totten Glacier from the 1960s. The authors present estimated historical glaciological changes and past ocean observations. This kind of work investigating the history of glaciological and oceanographic changes is valuable. I recommend that this manuscript for published in Nature Communications after minor revision. As I am an oceanographer, I only comment on the ocean part of this work.

Minor comment

“Therefore, the mCDW-induced basal melting process is responsible for the acceleration during our study period from 1963 to 1989 and further for the long-term acceleration trend of nearly 6 decades in the grounding line region of the Totten Glacier (Fig. 2C).” As there is no data between 1963-1989, I do not think authors can make a statement that mCDW induced basal melting “is” responsible. I think they should say “is likely” responsible. I think this sentence is too strong.

Response:

Thanks for the comment. As suggested, we changed the sentence to “*Therefore, the mCDW induced basal melting process **is likely** responsible for the acceleration during our study period from 1963 to 1989 and further for the long-term acceleration trend of nearly 6 decades in the grounding line region of the Totten Glacier (Fig. 2C).*”

Reviewer #2 (Remarks to the Author):

I have read through the updated manuscript, and the responses to my own comments on the initial draft of the manuscript. I am satisfied that my comments have been addressed sufficiently, particularly with regards to clarifying the spatial variation in the role of ocean melting vs ice shelf area, on ice velocity. The manuscript has also been significantly improved by the updates made to Figure 2, and the accompanying additional analysis.

From a scientific standpoint, I can now recommend publication once the few minor comments below have been addressed:

Minor comment

L36 – Should read ‘The Antarctic Ice Sheet’.

Response:

Changed to “*The Antarctic Ice Sheet*”.

L186 – full stop before thereafter.

Response:

Changed accordingly.

L252 – ‘primarily’ is perhaps a better word than ‘dominantly’.

Response:

Changed from “*dominantly*” to “*primarily*”.

L315 – note that while Box 2 is in the grounding line region, it is still located on floating ice, which means any velocity variability will not directly increase ice discharge. This sentence should be rephrased to highlight that this process can indirectly lead to increased ice discharge. Change throughout as necessary.

Note: line numbers refer to the non-tracked changes version of the updated manuscript.

Response:

Thanks for the comment. We changed the sentence to “*The acceleration in ice shelf front for 1973-1989 is attributed to the large calving front retreat, while that in the grounding line region is mainly caused by the mCDW-induced basal melting, indirectly leading to an increase in ice discharge.*”

We also changed a similar sentence in Line 200 (clean version of the manuscript).